# Extraction of accurate cytoskeletal actin velocity distributions from noisy measurements

Cayla M. Miller [1], Elgin Korkmazhan [2] & Alexander R. Dunn [1,3✉]

Dynamic remodeling of the actin cytoskeleton is essential for many cellular processes. Tracking the movement of individual actin filaments can in principle shed light on how this complex behavior arises at the molecular level. However, the information that can be extracted from these measurements is often limited by low signal-to-noise ratios. We developed a Bayesian statistical approach to estimate true, underlying velocity distributions from the tracks of individual actin-associated fluorophores with quantified localization uncertainties. We found that the motion of filamentous (F)-actin in fibroblasts and endothelial cells was better described by a statistical jump process than by models in which filaments undergo continuous, diffusive movement. In particular, a model with exponentially distributed jump length- and time-scales recapitulated actin filament velocity distributions measured for the cell cortex, integrin-based adhesions, and stress fibers, suggesting that a common physical model can potentially describe actin filament dynamics in a variety of cellular contexts.

[1] Department of Chemical Engineering, Stanford University, Stanford, CA, USA. [2] Biophysics Program, Stanford University, Stanford, CA, USA. [3] Stanford Cardiovascular Institute, Stanford University School of Medicine, Stanford, CA, USA. ✉email: alex.dunn@stanford.edu

The actin cytoskeleton is both dynamic and mechanically robust, allowing it to simultaneously define cell shape while facilitating membrane protrusion and cell migration. These roles require cytoskeletal assemblies with distinct physical properties[1–4]. Two such examples are the actin cortex, a thin contractile mesh immediately beneath the cell membrane that helps to maintain cell shape[5], and stress fibers, bundles of crosslinked actin and myosin that transmit force to the extra-cellular matrix (ECM) through focal adhesions[6]. These and other F-actin based structures generate force through a combination of pushing forces resulting from F-actin polymerization and pulling forces generated by motor proteins, most prominently non-muscle myosin II. Imbalances in these push-pull forces drive localized F-actin flows, which thus report on the local dynamics of cytoskeletal remodeling and cellular force generation[7].

Experiments quantifying local F-actin flow velocities in the leading edge of migrating cells have yielded key insights into how cell shape, migration, and force transmission arise at the mole-cular level. Early work examining F-actin dynamics noted rapid actin turnover and rearward flow from the cell edge toward the cell body in the lamellipodia of both keratocytes and fibroblasts[8,9], a phenomenon termed retrograde flow that has since been observed and characterized in many cell types[2,3,10–13]. Speckle microscopy established the lamellipodia and lamella as distinct regions with unique F-actin retrograde flow velocities and polymerization kinetics, pointing to specialized roles in ECM exploration and persistent advancement of the cell's leading edge, respectively[2,14]. Kymograph-based analysis of F-actin velocities in fish keratocytes demonstrated that myosin II inhibition can speed or slow retrograde flow, depending on the force balance in an individual cell[15]. Multiple studies have also shown that actin flows are slowed locally over integrin-based cellular adhesion com-plexes, termed focal adhesions, a phenomenon that reflects localized traction generation[16–19]. These and other studies sup-port a consensus understanding, termed the molecular clutch model, in which rearward flow of F-actin is coupled by frictional slippage to focal adhesions in order to exert traction on the cell's surroundings[16,20–22]. Fewer studies have focused on actin dynamics under the cell body, where directed, vectoral F-actin flow is less apparent[23,24]. This knowledge gap is potentially important, as the lamellipodia and lamella constitute a small fraction of the cell's surface in most cell types.

A separate body of work has used biophysical measurements to infer the mechanical properties of the cytoskeleton. Experiments that used magnetic beads attached to the cell's exterior to exert indirect forces on the cell cytoskeleton support a model in which cells, and by extension the cytoskeleton, can be described as soft, glassy materials close to the glass transition point, or else as gels[25–27]. In these models, actin filaments form a crosslinked network that rearranges over a broad range of timescales. This general view is supported by a variety of measurements, including optical trap experiments[28], atomic force microscopy measure-ments, and other techniques (reviewed in ref. [29]).

The large majority of studies characterizing the cytoskeleton have quantified bulk, or averaged, F-actin dynamics as opposed to those of individual filaments. Cell rheological measurements, as discussed above, necessarily average over large numbers of fila-ments. Similarly, the large majority of F-actin tracking mea-surements, such as those supporting the molecular clutch model, employ speckle microscopy, in which F-actin is labeled at a density such that puncta typically comprising several fluor-ophores are tracked. This technique, though powerful, does not straightforwardly report on the motion of individual fluor-ophores, and hence individual filaments[30]. A few studies have used sufficiently low concentrations of F-actin fiducials to track single fluorophores[19,31,32]. However, where these data have been

analyzed to yield information on the dynamics of individual filaments, the focus has remained on a relatively small number of filaments near the cell edge[19].

Recent results from our laboratory implied that actin filaments attached to integrin-based adhesions do not undergo continuous retrograde flow, but instead move with discontinuous stick-slip motion[33]. This picture contrasts with earlier molecular clutch models, which featured continuous, retrograde F-actin flow[34]. Discontinuous movement at the level of individual filaments would be consistent with a description of the cytoskeleton as a gel or glass[27,35]. However, to our knowledge no measurement of single-filament dynamics in living cells has been reported that differentiates between continuous flow and discontinuous slip-stick models of motion.

Here, we present a novel analysis that uses prior information about the localization error and Bayesian inference to estimate true underlying velocity distributions from noisy fiducial tracking data. Our measurements revealed details of distinct velocity distributions for F-actin populations inside and outside stress fibers, and also showed how the velocity distributions evolve over sub-minute timescales. At short timescales (<10 s), our analysis revealed the presence of stationary or slow moving F-actin, which was pre-viously undetectable due to measurement noise, along with a long tail of faster-moving filaments. At longer timescales (10 - 40 s), the velocity distribution became more narrowly distributed around a peak velocity. We propose one physical model that is consistent with these observations, namely a statistical jump process that describes the movement of individual actin filaments.

## Results

We quantified the motion of individual F-actin filaments in human foreskin fibroblasts (HFFs) by sparsely labeling with very low concentrations of SiR-actin, a probe whose fluorescence increases 100-fold when bound to F-actin. The use of SiR-actin provides several advantages: compatibility with many cell types without genetic engineering, live cell permeability without harsh electroporation or microinjection, and binding-specific fluores-cence resulting in low fluorescence background[36]. The cells were imaged using TIRF microscopy, limiting the fluorescence exci-tation field to the basal surface ( <200 nm) of the cell. The F-actin fiducials marked by SiR-actin were imaged every two seconds for a total of two minutes (Supplementary Movie 1). Time-lapse recordings revealed a small population of fast-moving, highly directed actin flowing retrograde at the cell edge (Fig. 1, a-b, pink, peach, and olive kymographs). While important in the context of lamellipodial extension, this population constitutes a small min-ority of the F-actin in most adherent cell types, and moreover has been intensively studied[2,3,10–15,17–19]. We therefore focused on F-actin populations under the cell body.

In general, F-actin in stress fibers moved slowly relative to lamellipodial F-actin (Fig. 1a, b, teal and blue kymographs), as previously shown[22,24]. We observed coherent motion of fiducials in the same stress fiber, and in many cases approximately constant velocities over the observation time window, suggestive of locally coordinated flows. We note that, while these measurements report on the motion of individual filaments, these motions are likely composed of a combination of individual filament motion and the coordinated movements of locally crosslinked structures. Finally, we observed a large pool of F-actin belonging to neither the slow-moving stress fiber population nor the fast-moving cell-edge population. This population represents the cortical actin network. It has been shown that this network is important for transmitting tension along the contour of the cell membrane[37], and exhibits a wide range of speeds in yeast (0–1.9 μm/s)[38]. However, the motion of this population is not well-characterized in mammalian cells.

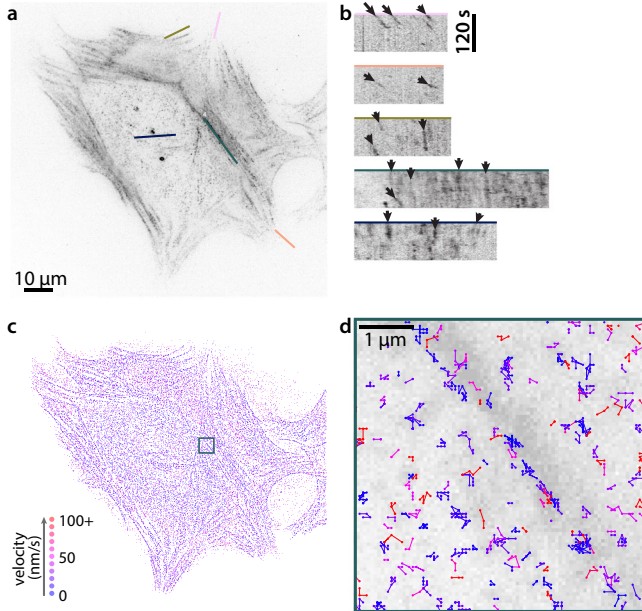

**Fig. 1 SiR-Actin labeling reveals variation in F-actin velocities in fibroblasts. a** An HFF labeled with 50 nM SiR-Actin. This labeling strategy was successfully repeated in live HFFs (5 experimental replicates), though SiR-Actin concentration was varied to achieve similar labeling densities (see Methods, Experimental Setup and Imaging). **b** Kymographs showing actin motion in the cell in **a** over 120 s, at the cell edge (pink and peach), closer to the cell body (olive), and in stress fibers (teal and blue). **c** Single particle tracking for the same cell over the same 120 s interval. The first point of each track is shown, color-coded by its apparent velocity. **d** Close up of the boxed region in **c**, showing all points of the tracks in and surrounding a stress fiber. The tracks are overlaid over the mean intensity projection of the time series. Scale bar is 1 μm.

**Accurate F-actin velocity distribution can be inferred from noisy measurements.** We used quantitative fluorescent speckle microscopy (QFSM)[39] to track SiR-actin puncta in all areas of the cell (Fig. 1c, d). These tracks were then filtered to identify those were likely to correspond to single fluorophores and processed to yield subpixel localizations for each frame (Methods, Actin Tracking Analysis). The resulting tracks showed that stress fibers contain slow moving, longer lasting tracks, consistent with the evidence from selected kymographs. However, velocities calculated from raw tracks are compromised by a potentially serious overestimation of the displacement between time steps (Fig. 2)[40]. For a fluorophore with a given localization error $\sigma_{xy}$ in the imaging plane which moves a true distance $s$ between frames, the measured distance $d$ is sampled from a noncentral $\chi$ distribution with 2 degrees of freedom, and $\langle d^2 \rangle = s^2 + 4\sigma_{xy}^2$ (Supplementary Note 1).

Consequently, for small displacements, the localization error overwhelms the measured distance. Figure 2a, b shows simulations of measured distances between two positions with a true distance $s = 5\sigma_{xy}$, as well as for a true distance of $s = 0$ (a stationary point) with $\sigma_{xy} = 1$. In both cases, the distribution of measured distances was in good agreement with the analytical solution given by a noncentral $\chi$ distribution. For the longer distance, the average measured distance is roughly correct, but for a stationary particle ($s = 0$), all measurements are greater than the true displacement of zero, giving a peaked distance distribution (with peak at $\sqrt{2}\sigma_{xy}$). For both true distances, the distribution of measured distances is broader than the (Dirac delta-function) true distribution.

In order to quantitatively characterize the localization error, we treated chemically fixed cells with SiR-actin and tracked the actin fiducials through time in the same way as our live-cell measurements. The resulting distribution of measured displacements was peaked at non-zero values, as predicted (Fig. 2c). However, this distribution is not well-fit by a single noncentral $\chi$ with $s = 0$, due to the varying localization errors of individual puncta. We are able to capture this variation by a mixture of two noncentral $\chi$ distributions, giving three fit parameters: $\sigma_{xy1}$ and $\sigma_{xy2}$, the $\sigma$s contributed from each of the two noncentral $\chi$ distributions, and $f_1$, the relative weighting of the noncentral chi with the smaller $\sigma$. Here and hereafter we report maximum likelihood estimates for fit parameters; MLE allows us to maximally leverage the information in tracks ($n = \sim70,000-360,000$ displacements in fixed cells from each experimental dataset), rather than fitting to the binned histogram. On average across all experiments, the best fit parameters were $\sigma_{xy1} = 23$ nm, $\sigma_{xy2} = 50$ nm, and $f_1 = 0.57$, defining the localization error in our system (Fig. 2c, Supplementary Fig. 1).

With this information, we can consider more complex cases, where the true distance, $s$, is not constant, but is a random variable sampled from some distribution, $f_s(s)$. In this case, the distribution of measured displacements depends on two factors: the probability distribution of the true distances, $f_s(s)$, and the conditional probability, $f_d(d|s)$ of measuring a displacement, $d$, given a true underlying distance, $s$. We use Bayes' theorem to infer the true displacement distribution from the distribution of measured displacements:

$$f_d(d) = \int_0^\infty f_d(d|s = S)f_s(S)dS \qquad (1)$$

where $f_d(d|s = S) \sim \chi_{NC}(S, \sigma_{xy})$, where $\chi_{NC}$ is the noncentral $\chi$ distribution. In our case, to capture the variability in localization errors $f_d(d|s = S) \sim f_1 \cdot \chi_{NC}(S, \sigma_{xy1}) + (1 - f_1) \cdot \chi_{NC}(S, \sigma_{xy2})$. Therefore, for a measured distribution of distances and quantified localization error, one can infer the distribution of true distances. It is important to note that the fitting is not model-agnostic, and requires an assumed functional form for $f_s(s)$. However, the fitting can be quickly solved numerically, and fitting to various distributions is trivial.

As an illustrative example, we fit simulated F-actin track data with true displacements drawn from a Weibull distribution, with Gaussian localization error (Fig. 2d–f). The Weibull distribution yields either an exponential or peaked shape, depending on the choice of parameters, and thus is capable of capturing a wide range of physical behaviors at a phenomenological level. For a case where average displacement, variance, and localization error are all the same magnitude, (Weibull shape = 1.5, scale = $\sigma_{xy}$), the true and measured distances are shown together in Fig. 2e. We fit the simulated distribution in (e) to Eq. (1), with $f_s \sim Wbl(k, \lambda)$, and the localization error fixed. The fit matched the simulated data well (e), and the best fit parameters accurately captured the true distribution (f). We used a similar workflow to infer true distributions from our live cell F-actin displacements given the localization error fit parameters from our fixed cell measurement (Fig. 2g).

**Fit velocity distributions show distinct physical properties for stress fibers and cortical actin.** Many previous F-actin tracking measurements have avoided the high errors associated with slow actin speeds by focusing on regions of the cell where movements are fast, by filtering out nearly-stationary particles in a population, or by imaging over longer timescales (minutes to tens of minutes), where the displacements become larger. In our study,

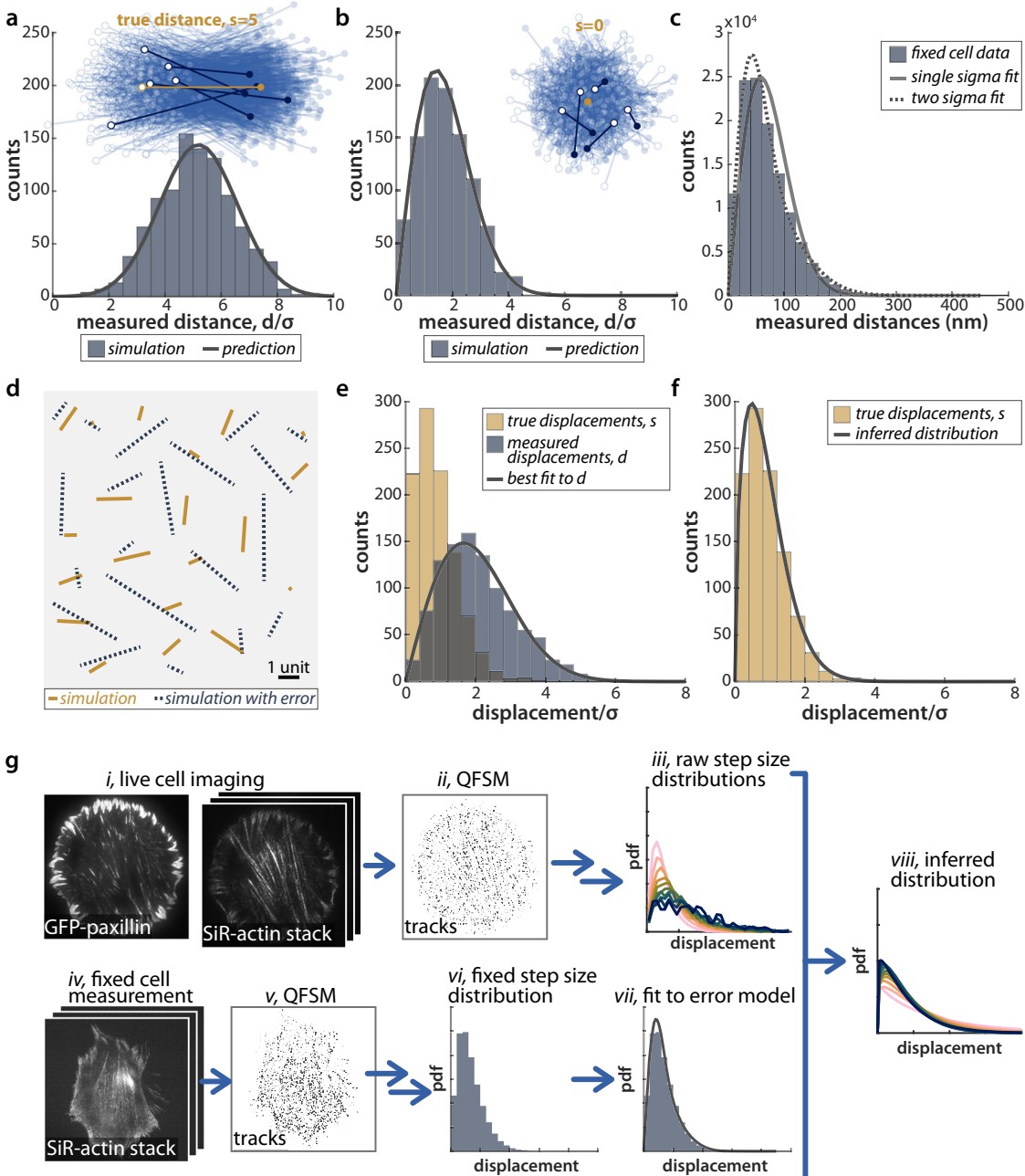

**Fig. 2 Bayesian fitting to reconstruct a distribution of true distances that are otherwise obfuscated by localization error. a**, **b** Simulated pairs of points sampled around two true locations separated by a distance $s = 5$ (**a**) or $s = 0$ (**b**), with localization error $\sigma_{xy} = 1$. The first point of each distance measurement is marked by an open circle, and the second point by a closed circle. Five representative pairs are shown in heavier weight. The histograms of measured distances and the predicted distribution of measured distances given by a noncentral $\chi$ are shown below. **c** The distribution of displacements measured in fixed cells (measured over a 2 s increment) fit to a noncentral $\chi$ distribution with $s = 0$, or the sum of two noncentral $\chi$ distributions, each with $s = 0$ and differing values of $\sigma$. n = 10 cells. **d** Simulated frame-to-frame motion of 20 simulated particles, showing both the true displacements, drawn from a Weibull distribution (shape = 1.5, scale = $\sigma_{xy} = 1$) and the error-laden measured displacements. **e** The distribution of true displacements and measured distances for 1000 such particles, as well as the best fit to the measured distribution. **f** The resulting inferred true distribution from the fit in **e** overlaid with the true distribution of distances. **g** This method is applied to measurements of SiR-actin in cells: i, iv. Images of both GFP-paxillin-marked adhesions and SiR-actin are acquired on a TIRF microscope. ii, v. QFSM[39] is used to identify puncta and tracks through time. iii, vi. Tracks are filtered and sub-pixel localized to generate a distribution of displacements across the cell. vii, viii. Using the fixed cell data as a measure of the localization error, we can infer the true displacement distribution from live-cell measurements. Source data for **a**–**f** are provided in the Source Data file.

we used non-subsequent frames to measure displacements over a number of timescales ranging from 2 to 40 s with a constant exposure of 0.3 s per frame. During our analysis, we sorted F-actin fiducials into four mutually-exclusive populations based on their location within or outside stress fibers, and over or not

over focal adhesions. For all of these subpopulations the peak of the distribution moved rightward and the distribution broadened at longer timescales (Fig. 3a). However, because of the contribution from localization error, the shapes and magnitudes of these distributions cannot be directly interpreted.

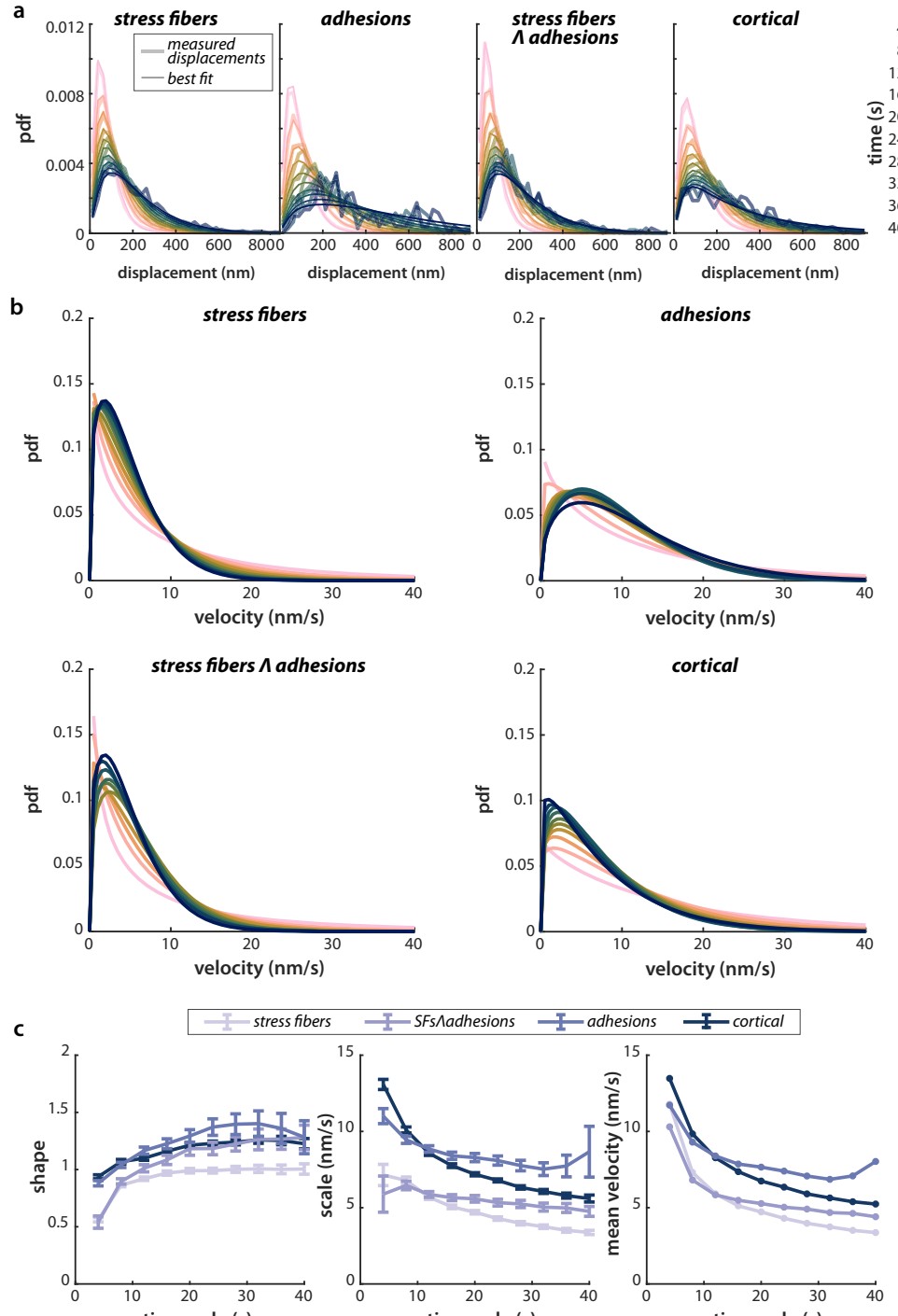

**Fig. 3 Inferred distributions of F-actin velocities show distinct behavior inside vs. outside SFs. a** Puncta were automatically classified into subpopulations, corresponding to the four mutually-exclusive combinations of (±) stress fibers and (±) adhesions: the cortical population is taken to be F-actin outside both stress fibers and adhesions. Displacements for each subpopulation, along with best fits, are shown for varying time intervals. The distribution of displacements is calculated for timescales ranging from 2 s to 40 s, in 2 s intervals. Only every other time interval is shown for clarity. **b** The inferred velocity distributions corresponding to the measured distance distributions in **a**, assuming that underlying displacements follow a Weibull distribution. Example data and fits in **a**, **b** are for representative from one experiment, $n = 9$ cells. **c** Left, center: Fit parameters for the Weibull distribution fits. Error bars show the 95% confidence interval for each parameter, determined by normal approximation (see Methods, Model Fitting). Right: the average velocity for each population and timescale, calculated from the best-fit Weibull distribution. The best fit parameters in (**c**) represent the average over four experimental replicates, comprising 32 cells in total. Source data for all plots are provided in the Source Data file.

We inferred true velocity distributions by fitting our live cell measurements using Eq. (1), as described in the previous section and in Supplementary Note 2. There are many functional forms to choose from for $f_s(s)$, but for parsimony, we began with the simplest: the exponential, Gaussian, and Rayleigh distributions. These reflect actin movements driven by a single-step Poisson process, averaged movements described by the central limit theorem, and purely diffusive 2D motion, respectively. An

exponential distribution fit F-actin displacements in stress fibers reasonably well over all timescales examined here (2–40 s), but did not fit the cortical actin population at longer timescales, Supplementary Fig. 2). Because our distance data are constrained to be positive, we fit to a folded Gaussian, rather than true Gaussian distribution. This distribution fit most of the data reasonably well but failed to capture the tails of the distribution (Supplementary Fig. 3). If the true distribution were governed by a 2D, purely diffusive process, we would expect the distribution of step sizes to follow a Rayleigh distribution. This distribution largely failed to capture the behavior of the F-actin tracks in both stress fibers and the cortex (Supplementary Fig. 4).

We next tested the Weibull distribution, a two-parameter model that interpolates between the exponential (Poisson process) and Rayleigh (diffusive) distributions:

$$f(s) = \frac{k}{\lambda} \left( \frac{s}{\lambda} \right)^{k-1} e^{(s/\lambda)^k}, \qquad (2)$$

where $k$ is the shape parameter, and $\lambda$ is the scale parameter. This distribution fit the data well for all four populations over all timescales. Comparing the likelihood ratios of these fits with the next best fitting, the folded Gaussian, revealed that the best-fit Weibull is $10^2$–$10^{76}$ times more likely than the best-fit folded Gaussian for each population (Supplementary Table 1).

We scaled the fit distance distributions by the observation time to generate velocity distributions for all timescales (Fig. 3b). This affects only the scale parameter, not the shape parameter, of the distribution. The shape parameter, $k$, from these fits showed a relatively constant value around 1, indicating that the behavior on these timescales is more similar to exponential-like behavior ($k = 1$) than Rayleigh-like behavior ($k = 2$) (Fig. 3c). Physically, F-actin motion was thus more like Poissonian motion than diffusive motion. Interestingly, average velocities varied inversely with the timescale of the measurement, an effect that was not captured by a simple Poisson process (see *F-actin velocity distribution fits can estimate physical parameters of underlying motion* below).

To determine to what extent the variation we observed is a result of pooling together F-actin tracks from several different cells for analysis, we examined the velocity distributions for individual cells. We found relatively little cell-to-cell variability in velocity distributions in both the cortex and stress fibers (Supplementary Fig. 5a, b). Fits of the data from individual cells yielded fit parameters that showed the same trends as the population as a whole: in all cells, the stress fiber population was best fit by smaller shape and scale parameters than was the cortical population (Supplementary Fig. 5c, d).

As a separate comparison of the F-actin behavior in the stress fiber and cortical populations, we also fit the lifetimes of our speckles as described in Supplementary Note 4 and shown in Supplementary Fig. 11. From this analysis, we found the turnover rate to be almost twice as fast in the cortical population as in the stress fiber population (with average lifetimes of 18 and 34 s, respectively).

**F-actin velocity distribution fits can estimate physical parameters of underlying motion**. While the Weibull distribution yielded an excellent empirical description of our data, it provided limited physical insight into the mechanistic processes that drive F-actin motion. We therefore sought to develop a reductionist physical model of F-actin motion that could potentially account for our observations. As with all reductionist models, it likely fails to capture important aspects of cellular behavior (here, F-actin dynamics) that arise from the complex molecular milieu of the cell. Moreover, as with all such models, its correctness cannot be "proven," but only invalidated by experimental data.

Nevertheless, we view such models as useful in testing physical intuition, and in generating predictions that can potentially stimulate future experiments.

F-actin movement in the cell may plausibly arise from some combination of diffusive motion, bulk drift, and jump processes. We therefore interpreted our data by comparing two extremes: one in which F-actin moves solely by diffusion with drift (Fig. 4a), and one in which F-actin moves solely by a jump process (Fig. 4b, described in Supplementary Note 3). For a population which moves with some drift velocity in one direction and diffusion in two dimensions, the step size is described by a Rice distribution (mathematically equivalent to the noncentral $\chi$ distribution with two degrees of freedom, but parameterized in a way that is more intuitive in this context). Statistical jump processes are less often modeled in a biological context, and describe a system where particles undergo abrupt jumps in location, termed a position jump model, or else in velocity, termed a velocity jump model. Both position and velocity jump models yield descriptions of actin filament motion that are essentially identical at the spatiotemporal resolution of our experiment (Fig. 4b). However, the position jump model yields an analytically tractable model (see below). Although the motion of the F-actin cytoskeleton cannot be truly instantaneous, a position jump model provides a useful framework to approximate processes like the rapid slipping and sticking in the molecular clutch model[41], or those that might be expected from soft glass or gel descriptions of the cytoskeleton[25–27]. For the simplest jump motion process, we assume both the jump time and jump distance are exponentially distributed.

Consistent with our observations from the Weibull fits, the data were much better fit by the (Poisson-like) position jump process than by the model of diffusion with drift (Fig. 4c, d). In this simple formulation, any piece of the cytoskeleton moves some random distance after waiting a random time. At each time, a new distance and wait time are sampled. The distance and the time are both exponentially distributed, and fits to this model gave mean switching distances and timescales of ~5−25 nm and ~20−70 s, respectively (Fig. 4f), with stress fibers showing slightly longer but less frequent jumps than the other three populations.

A drawback to the 1D jump model is that, similar to the Weibull fits, it yields a timescale dependence in the best fit parameters, indicating that this model does not sufficiently account for the timescale evolution of our data. A number of factors may contribute to this timescale dependence: reversals in the direction of motion, F-actin motion that is not constrained to 1D, or nonzero velocities between jumps. Models incorporating these possibilities are not analytically tractable, but can be fit to the data using an iterative fitting approach combined with Monte Carlo simulations. Here we present the simplest model that adequately describes the data: 1D jumps with exponentially distributed jump distances and wait times as above, that switch direction with some probability $P_{switch}$ (Fig. 4b).

The 1D jump model with reversals successfully fit data from each F-actin population across timescales from 2 to 40 s using only 3 global parameters (Fig. 4e and Supplementary Fig. 6). The best fit time between jumps varied most: actin in stress fibers was best fit by average wait times of 17 s while that in cortical actin was only 5 s (Fig. 4g). The two populations over adhesions, with and without stress fibers, were best fit by intermediate timescales of 10 and 6 s, respectively. There was less variation in characteristic jump distances, with stress fibers yielding a best fit of 70 nm and the other three populations yielding jump distances of 45–51 nm. The resulting inferred velocity distributions are plotted in Supplementary Fig. 7. The best fit jump distance and times are similar to those from the analytical solution for the 1-D position jump without reversals, which requires far less computational time to fit. Interestingly, the reversal probability was lower over adhesions than for either cortical or stress fiber F-actin, an

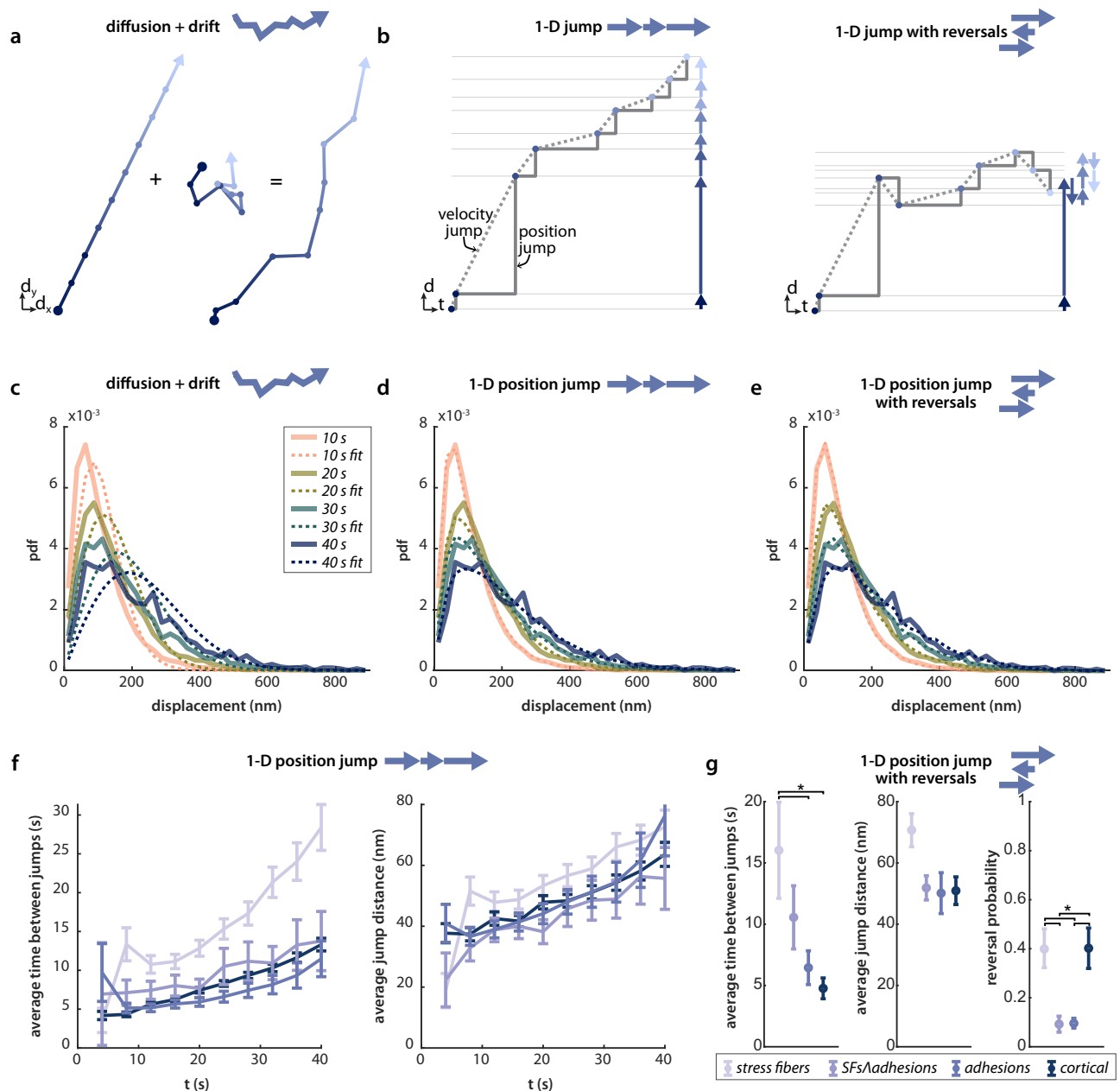

**Fig. 4 F-actin velocity distributions in HFFs are well-fit by a jump process model with exponentially distributed waiting times and jump distances.**
**a** Schematic of the diffusion with drift model. Persistent drift (left) is coupled with diffusion (center), leading to the motion shown on the right. **b** Schematic of the jump model, both with (right) and without (left) direction reversals. The one-dimensional motion is shown both in two-dimensional distance-time space and in one-dimensional space (arrows). The solid lines represent the motion of a position jump model, while the dashed lines represent the analogous velocity jump model. **c**–**e** The best fits to the stress fiber population data by each of the models in **a** and **b**: diffusion with drift (**c**), 1-D position jump (**d**), and 1-D position jump with reversals (**e**). The fits are shown for four timescales and are for representative data from one experiment (n = 9 cells). **f** The best fit parameters for the 1-D jump process, for all timescales measured, averaged over four experiments. Error bars show 95% confidence intervals on the fit parameters, propagated across the four datasets (see Methods, Model Fitting for details). **g** The best fit parameters for the 1-D jump process with reversals, fit to all timescales (2–40 s), averaged over four experiments, n = 32 total cells. Error bars show standard error of the mean of the fit parameters from four independent datasets. Groups were compared with one-way ANOVA and asterisks denote comparisons with p-values < 0.05. P-values for comparisons between all groups are tabulated in Supplementary Table 3. Source data for **c**–**g** are provided in the Source Data file.

observation that is consistent with the recruitment of F-actin nucleators to integrin-based adhesions (see Discussion).

The jump models introduce a discontinuous point at zero, given that there is finite probability that a particle does not jump at all during an observation time (Supplementary Fig. 7). The fraction of the population that is motionless is set only by the characteristic timescale of the jumps and the timescale of

observation, and is independent of the characteristic jump length. This probability is high at short timescales (89% for stress fibers and 67% for cortical at the 2 s timescale) and diminishes at longer timescales (9% for stress fibers and 0.03% for cortical at the 40 s timescale). This recovered population from fitting is consistent with the observation that many puncta appeared to be stationary by eye or by kymograph, particularly in stress fibers.

We also tracked F-actin fiducials in human umbilical vein epithelial cells (HUVECs), which when plated on glass formed numerous, well aligned stress fibers. F-actin in this cell type was best fit by a jump process with best fit parameters similar to HFFs, but somewhat shorter waiting times between jumps in the stress fiber population (Supplementary Fig. 8 and Movie 3).

We next examined how the motion of individual actin filaments varied in response to perturbations known to affect cytoskeletal dynamics at a more global level. The kinetics of biological macromolecules are temperature sensitive, with rates that typically change by a factor of 2–4 with every 10 °C change in temperature[42,43]. When we quantified F-actin dynamics in HFFs imaged at a lower temperature (26 °C), we found that F-actin motion was best fit by similar jump distances, but less frequent jumps when compared to measurements at 37 °C (Supplementary Fig. 9). We also explored the effect of decreasing the overall abundance of F-actin by treating cells with latrunculin A, which depolymerizes F-actin and sequesters G-actin monomers[44]. F-actin velocities in HFFs treated with 200 nM latrunculin A were best fit by increased jump distances and waiting times between jumps, as well as an increased probability of reversal, suggestive of a disruption of the cytoskeletal network (Supplementary Fig. 10). The effects of treatment with 25 μM blebbistatin, a concentration chosen to preserve the presence of some stress fibers and adhesions, were more muted (Supplementary Fig. 10).

## Discussion

We have described and applied a method to infer velocity distributions for slow single particle tracks in living cells. While recent microscopy techniques have allowed precise measurements with single nanometer accuracy in vitro[45], these advances are often difficult to apply to commonly used in vivo fluorescent probes. Because our technique does not require high precision to reconstruct a population with slow-moving molecules, we anticipate that it may be useful in the analysis of other particle tracking measurements that face limitations imposed by low signal-to-noise ratios. The approach we outline here is also applicable to the statistical description of noisy distance measurements, for example in the context of two-color STORM/PALM microscopy experiments, and may thus be generally useful in interpreting superresolution data.

While F-actin motion often has the visual effect of steady flow at the bulk level, it was unknown how the motion of individual filaments might be represented statistically. Like diffusion of particles down a concentration gradient, which has the visual effect of steady unidirectional movement, the paths of individual particles need not demonstrate steady forward motion. Here, we measured the motion of single puncta, and found these movements to be inconsistent with a model in which individual particles move at a constant velocity (Fig. 4, Supplementary Fig. 16). Our findings are however consistent with previous studies whose results implied heterogeneity in the velocities of individual actin filaments[19], and in particular that a subset of F-actin filaments should have zero actin velocities[33,46]. Thus, single-molecule and bulk measurements are not inconsistent: relatively rapid and heterogeneous movements at the single-filament level, averaged over many filaments and multiple filament lifetimes, are expected to yield the ensemble-level movements seen previously.

Though a number of models can be fit to the data, goodness of fit allowed us to eliminate several simple distributions (exponential, Gaussian, Rayleigh), which constrains possible models of actin motion and force transmission. We find that of the analytical distributions we tested, the Weibull fit best and was able to capture a range of actin behaviors. We used numerical simulations to fit the same data to a mechanistic model of actin motion. Surprisingly, jump-type models fit the data significantly better than the more common, continuous diffusion with drift framework for all populations, despite the diversity of organization and function of these structures. While the analytically tractable 1D position jump was sufficient to fit our measured displacement distributions for a given timescale, the inclusion of stochastic directional reversals was sufficient to recapitulate our data across timescales with only 3 fit parameters. The ability to describe a variety of F-actin populations with a relatively simple model suggests that the emergent physical properties of the cytoskeleton may be understandable within a common physical framework despite the considerable complexity of the cytoskeleton at the molecular scale.

While we don't observe obvious jump-like movements in our videos, the average jump distance for most populations and timescales is roughly 50–70 nm, close to our localization precision and to our pixel size of 64 nm. Given the exponential distribution of jump distances, the most common jump sizes would be expected to be close to zero, and a majority of jumps would be less than one pixel. Physically, F-actin motion is unlikely to be truly instantaneous, as in this formulation of a position jump process but could be much faster than our frame rate of 2 s.

As in other studies, we noted fast, inward-directed fiducial motion at the cell's periphery, consistent with retrograde flow in lamellipodia, but these make up a small number of our tracks (Supplementary Movie 1). It is plausible that F-actin motion in the lamellipodia is dominated by advective flow, i.e., drift, with respect to the laboratory rest frame, as in previous studies that focused on lamellipodial F-actin dynamics. Whether filaments additionally undergo jump-type motion within flowing lamellipodial/lamellar actin remains to be determined. Perhaps relatedly, the best fit jump model in regions over adhesions, which are often close to the cell edge, yielded fairly low probabilities of switching direction (~10%). It is possible that this low probability of reversal may reflect F-actin polymerization at adhesions, for example by by formins or VASP[47,48].

Although we emphasize the provisionality of the jump model, it is interesting to consider its physical interpretation. In the context of a position jump model, one interpretation is that, on this timescale, F-actin exists in a crosslinked network that is close to mechanical equilibrium. Disruption of this equilibrium by either crosslinker unbinding or force generation by myosin occurs in discrete jumps reflecting the intrinsically quantized, molecular nature of these processes. In this interpretation, the characteristic lengths, waiting times, and reversal probability of the actin cortex and stress fibers reflect the emergent properties of each structure. This picture is consistent with the structural arrest and rearrangements that would be expected from descriptions of the cytoskeleton as a soft glass or gel[25–27].

Variations in the local density of actin crosslinkers and binding proteins may contribute to population changes in jump distance, as evidenced by the increase in jump size observed in cells treated with latrunculin A. In this regard it is intriguing that the magnitude of jump distances (~50 nm) is similar to the actin network mesh size (10–100s of nanometers)[49]. Waiting times between jumps may reflect the balance of kinetics between force-generating myosin motors and structure-stabilizing actin crosslinkers. Times that span roughly 5–25 s are commensurate with the timescale of F-actin crosslinker rearrangement inferred in a recent study from our laboratory[33]. The longer time between jumps in the stress fiber population may likewise reflect the greater stability of these structures, which have numerous cross-linking proteins in addition to force-generating myosins. At a lower temperature (26 °C), which is expected to slow molecular-level kinetics, the best fit characteristic time between jumps is increased, but both cortical and stress

fiber populations exhibit similar average jump distances relative to those measured at 37 °C. Last, the probability of switching may reflect the inherent directionality of local forces. For example, the switching probability was lowest over focal adhesions, where there is consistent inward-oriented traction forces as well as local actin polymerization, both of which might be expected to yield directional F-actin movement.

It is perhaps surprising that the jump model, or any model, fits the data given the compositional complexity of the cytoskeleton. It is likely that the best fit characteristic times, distances and reversal probabilities emerge from a distribution of timescales (and possibly length scales) that might be expected from the myriad of actin-binding proteins with varied kinetics, and that may be characterized by localized heterogeneities in space and time. Whether these emergent parameters arise in some limiting manner (e.g., due to central limit or extreme value theorems) or instead have a more complex physical origin remains to be determined. Alternative jump model formulations may also be useful in describing cytoskeletal motion. For example, a velocity jump model, in which particles move at constant velocity before "jumping" to a new velocity, is functionally similar to the position jump model at the spatiotemporal resolution of our measurement, as both models capture local heterogeneity of F-actin velocities in both space and time.

In summary, while any number of models may also account for our observations, we find that a jump model is sufficient to describe the motion of four cellular F-actin populations in the cell types examined here. Future work can help to elucidate the underlying molecular mechanisms that set the observed velocity distributions and more particularly, the biophysical origins of the distances and timescales implied by the jump model. It will likewise be of interest to determine how these properties vary across cell types, lending insight into how cells tune the molecular-scale dynamics of the cytoskeleton to fulfill different biological functions.

## Methods

**Cell culture**. HFF cells CCD-1070Sk (ATCC CRL-2091) were cultured in DMEM high-glucose medium (Gibco, catalog no. 21063-029) in the absence of phenol red and supplemented with 10% fetal bovine serum (Corning 35-011-CV), sodium pyruvate (1 mM, Gibco 11360070), MEM nonessential amino acids (1×; Gibco 11140050), and penicillin/streptomycin (100 U/ml; Gibco 15140122). HFFs stably expressing eGFP-paxillin (fused at the C-terminus) were prepared as previously described[50]. HUVEC cells (Lonza C2519A) were cultured in EGM-2 basal medium (Lonza CC-3156) with the addition of EGM-2 MV Bullet Kit (Lonza CC-4147), with the exception of gentamicin. Instead, penicillin-streptomycin (Gibco 15140122) is added to a concentration of 100 U/ml. All cells were grown at 37 °C with 5% CO2.

**Experimental setup and imaging**. Halo-PEG coverslips were prepared as previously[33]. Coverslips with coverwell chambers were functionalized with an RGD-presenting protein force sensor, here without fluorescent dyes, as described previously[33]. In brief, Halo-ligand functionalized coverslips were incubated at room temperature for 30 min with 100 nM unlabeled RGD-presenting sensor, which contains a HaloTag for attachment. The coverslip chambers were then rinsed with PBS before cells were seeded and allowed to spread for at least 1 h. After a 9 min incubation with 20–80 nM SiR-Actin (Cytoskeleton Inc., CY-SC001), the sample was incubated with Prolong Live Antifade Reagent (Invitrogen P36975) for 1 h. Optimal SiR-Actin concentration varied from batch to batch of the SiR-Actin reagent, but was chosen such that stress fibers could be identified in early frames, while still allowing for individual speckle tracking. For each cell, a 100 ms exposure of the GFP-paxillin channel (for masking) was first acquired, followed by a 60 frame sequence in the far red channel (SiR-Actin) with 300 ms exposures taken every 2 s. During experiments, temperature was maintained using an objective heater set to 37 °C, and the sample was stored at 37 °C before imaging. For the experiment in Supplementary Fig. 9, the objective temperature was instead set to 26 °C, and the sample was given time to cool from 37 °C before imaging. For fixed cell data, cells were allowed to spread on functionalized coverslips and then fixed in 4% paraformaldehyde for 15 min at room temperature. After rinsing, the cells were treated with SiR-Actin and Prolong reagent as above.

**Actin tracking analysis**. Speckles and tracks were first identified using QFSM software made available by the Danuser lab[39]. Next, the data were denoised using noise2void[51], and puncta previously identified by QFSM were fit to subpixel positions by Gaussian fitting of the denoised images. A small sample of representative tracks is shown in Supplementary Fig. 16. Any puncta which could not be successfully fit to a Gaussian were discarded. Unsuccessful fitting was defined as a fit center >2 pixels away from the original center position, or a fit standard deviation that did not lie within a factor of 1.5 of $0.25\lambda/NA$ following denoising, where $\lambda$, the wavelength of light was ~ 675 nm, and the NA of the objective was 1.49 (Supplementary Movie 2). Drift was calculated for each movie using the drift estimation module of NanoJ[52]. The calculated drift values in $x$ and $y$ were then subtracted from the positions of each point in the track before calculating frame-to-frame displacements.

The total cell area was masked using a mean threshold of the GFP-paxillin image, where cytoplasmic paxillin signal was used to segment the cell. After dilating and eroding to fill small holes, the edge of this mask was smoothed by fitting to a cubic spline curve. Adhesions were masked by an Otsu threshold of the same GFP-paxillin image after background subtraction to remove the diffuse cytoplasmic signal. Actin stress fibers were masked as the brightest 2% of pixels of a time-series projection of the actin tracks. Finally, the cytosol was taken as regions within the cell mask which are excluded from both the actin mask and the paxillin mask. Tracks were sorted into subcellular populations according to the location of the first point of the track.

To derive an empirical estimate of the localization error, we acquired SiR-actin tracks in paraformaldehyde fixed cells exactly as described above and used the displacements of these tracks to determine the measurement noise. The displacements of these fixed cell tracks were fit to a mixture of two noncentral $\chi$ distributions (described in Supplementary Note 1) with $s = 0$ by MLE. For each day's experiments, a corresponding fixed cell dataset was collected, tracked, and fitted, though variation in fixed cell samples across datasets was low. (Standard errors of the means (SEMs) across 11 fixed cell datasets for the two fit sigma values were 1.2 and 1.3 nm, while SEM for the fractional weighting of the mixture was 0.03.)

**Model fitting**. Frame-to-frame displacements were calculated for varying frame intervals, from 1 frame (2 s) to 20 frames (40 s). The displacement distribution for each interval was first fit by least-squares to quickly estimate the best fit parameters, and these parameters were used to seed the MLE fit to equation (1), which was numerically integrated as described in Supplementary Note 2. The resulting distributions were rescaled by the time interval to give velocity distributions. The error bars on fit parameters represent 95% confidence intervals on the parameters from normal approximation. In short, these ranges were calculated from a normal distribution with means given by the best fit parameters and variances given by the diagonal elements of the asymptotic covariance matrix estimated by MATLAB function mlecov. Mean fit parameters over multiple experimental replicates are presented in Figs. 3c and 4f, with error bars propagated from the ranges on parameters from individual experiments.

For the 1D position jump model with reversals, we used Monte Carlo simulations to iteratively fit the data. First, 10,000 particles were simulated using a Gillespie algorithm for at least 40 s. Each particle waits an exponentially distributed time, and then moves an exponentially distributed distance. There was a constant probability at each step of reversing direction from the previous step. The location of each particle at various observation times (0, 2, 4, ..., 38, and 40 s) was then determined, and Gaussian noise was added to the $x$ and $y$ position of each localization, according to the parameters from the fixed cell fits. For example, for fixed cell fit parameters of $f_1 = 0.6$, $\sigma_1 = 25$ nm, and $\sigma_2 = 50$ nm, the noise was sampled from a Gaussian distribution with $\sigma = 25$ nm with 60% probability and from a Gaussian with $\sigma = 50$ nm with 40% probability.

Displacement distributions were then calculated for twenty timescales (2, 4, ..., 38, and 40 s) from the simulated particles. Within each estimated pdf, bins with zero counts were set to $1/(N \cdot w)$, where $N$ is the number of particles in the distribution and $w = 30$ nm is the width of the bin; this prevents infinite likelihoods but overestimates rare events, as it represents an approximate upper bound on the expected value of that bin. To calculate likelihoods for our experimentally measured displacements, probability densities were linearly interpolated from the two closest points. To keep computational times reasonable, we calculated likelihoods from a randomly sampled subset of our data: for each of the twenty timescales, if there were more than 1000 displacements measured, a random sample of 1000 were chosen, resulting in ~20,000 observations included in the fit sample. Negative log likelihoods were minimized using MATLAB's genetic algorithm global optimization function. Due to the rough nature of the fit landscape (in part due to the stochasticity of the simulations), ten optimizations were completed for each dataset and the best fit (highest likelihood) was chosen for each. For each population, best fit parameters from different experimental replicates were averaged and the standard error of the mean on each parameter is presented (Fig. 4g).

The resulting velocity distributions from each fit parameter were generated using similar simulations, but without the addition of Gaussian noise to each localization. The discrete probability of zero displacement was calculated analytically as $P(s = 0) = e^{-\mu\tau}$, where $\mu$ is the average jump rate and $\tau$ is the observation timescale. For values > 0, probability distributions were calculated numerically from simulations with 100,000 particles.

For the drug perturbations presented in Supplementary Fig. 10, 95% confidence intervals were estimated using the profile likelihood method as described in ref. [53]. In short, each parameter was systematically varied in turn above and below its best fit value. At each sampled value, the other two parameters were again optimized using MATLAB's genetic algorithm, and the log-likelihood of this optimized set of parameters was recorded. These log-likelihood values were then fit to a parabola as a function of the varied parameter with its minimum fixed at the log-likelihood of the best fit parameter. The 95% confidence intervals were then estimated by the domain over which the parabolic fit to the $\log_{10}$-likelihoods rises by 0.8338 from the minimum.

As an additional test of the robustness of our measurement, we measured the effects of photobleaching and speckle density (Supplementary Fig. 12), variations in speckle brightness (Supplementary Fig. 13), spatial variation within the cell (Supplementary Fig. 14), and track lifetime (Supplementary Fig. 15 and Supplementary Table 2) on our measurements. While changes in some of these factors, in particular those which are correlated with localization error, caused shifts in the best-fit parameters, these changes were minimal, and trends in the data were preserved (Supplementary Note 5).

**Reporting summary**. Further information on research design is available in the Nature Research Reporting Summary linked to this article.

## Data availability
The microscopy data and corresponding tracked puncta generated in this study have been deposited in the Zenodo database under https://doi.org/10.5281/zenodo.6609641[54].

Source data for all graphs in both main text and supplementary figures are provided in the Source Data file. Source data are provided with this paper.

## Code availability
Sample code used to generate the results of this study has been deposited in the Zenodo database under https://doi.org/10.5281/zenodo.6609731[55].

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

## Acknowledgements

We thank S. Tan for providing RGD-presenting tension sensor and GFP-paxillin-expressing HFFs, C. Vasquez for assistance and advice with HUVEC handling, and V. Vachharajani, L. Owen, S. Tan, C. Hueschen, and B. Zhong for many useful discussions and feedback. Research reported in this publication was supported by grant R35-GM130332 to A.R.D. from the National Institutes of Health (NIH). The research of A.R.D. was supported in part by a Faculty Scholar Award from the Howard Hughes Medical Institute. C.M.M. was supported by Graduate Research Fellowships from the National Science Foundation (00039202) and the Stanford EDGE fellowship. E.K. acknowledges support from the Stanford Bio-X Fellowship. The contents of this publication are solely the responsibility of the authors and do not necessarily represent the official views of the NIH.

## Author contributions

C.M.M. performed experiments, processed the data, and performed the Bayesian modeling and analysis. E.K. and C.M.M. developed mathematical formulations for the physical models. C.M.M., E.K., and A.R.D. conceived the project and wrote the manuscript.

## Competing interests

The authors declare no competing interests.
