## [Peer Review File · Nature Communications]

Extraction of accurate cytoskeletal actin velocity distributions from noisy measurementsReviewers' Comments:

Reviewer #1:

Remarks to the Author:

This is a very interesting and clearly written study that investigates small and slow displacements of elements of actin network in various segments of the ventral actin network of an adherent cell - in the cortex, stress fibers, focal adhesions. The authors focus on very small displacements, the problem with which is that measurement error overwhelms true displacements. To overcome this obstacle, the authors use an elegant Bayesian method, which can predict probability distribution for true displacements, based on distribution for errors that is measured for fixed actin network, but this prediction depends on an assumed functional form of the true displacements probability distribution. The finding is that the so-called Weibull distribution describes the data well. Furthermore, the authors show that the best model that agrees with these data is discrete stochastic jumps of a few tens of nm size, every few seconds.

Potentially, these are formidable results. Here are some critical comments (some of them probably stem from my ignorance or misunderstanding):

1) Relevance for cell biology: the results are very curious, but I am struggling to understand what is the impact of this knowledge. OK, so there are small and slow discrete displacements of actin filaments... Does it affect cell movement? Actin network turnover? Perhaps, there are implications for actin network rheology? Other than very vague discussion that the results imply elastic properties of the network (which I don't find convincing anyway), I don't see anything else... The authors might want to get together with someone like Fred MacKintosh to think further.

2) There is a good discussion that the inferred displacements could be the result of crosslinking- and/or myosin-pulling events. The obvious suggestion: the study would be much more convincing, if the authors do relatively simple experiments: 1) use blebbistatin and caliculin to perturb contractility; 2) inhibit and overexpress alpha-actinin or other crosslinkers; 3) use latrunculin and jasplakinolide to change actin network. By the way, the characteristic displacements probably have nothing to do with helical structure of actin filaments, but very likely are the same order of magnitude as the actin mesh size - this also could be tested.

3) Why are the authors sure that what they observe is displacements of individual filaments, and not more complex network structures?

4) I could not figure out from the text: are the true displacements always isotropic, and their directions uncorrelated between consecutive movements? For the cortex, this is very possible, but I find this very puzzling for stress fibers, where there is constant slow flow along the fiber's length, and for adhesions, where the displacements are supposed to be biased in the direction of the center of the fiber on which end the adhesion is. Again, some myosin perturbations may give a clue here.

5) I find it extremely disturbing that the characteristic time between jumps grows with observation time. Clearly, the authors are also bothered by this, but the guess in the discussion that this has something to do with a more

stable actin population does not sound convincing to me. Speaking of which: the actin network is supposed to turn over on time scales similar to measurement times. First of all, the turnover time probably could be estimated by FRAP, or some other techniques. Second, were there discarded shorter trajectories? Basically, the issue of the turnover should be more carefully discussed. Relevant to that - there are likely some displacements away from the focal plane, no?

Reviewer #2:

Remarks to the Author:

The authors attempt to infer the hidden characteristics in the motion of actin filaments in live cells. To do so, they developed a method to statistically subtract the tracking error distribution from the observed distribution of F-actin speckle displacement in time series images. Although the scope of this paper may potentially be important to the research field, the paper lacks careful consideration of the mechanisms of error generation in object tracking for low signal to noise ratio (SNR) data.

The method developed in this paper uses a very simplified assumption in estimating the distribution of tracking errors; that is "measured coordinates are Gaussian distributed around the true location ..." in Supplementary Information p2. This is overwhelmingly too simplified. As found in Figure 2 of Nature Methods 11, 281, 2014 and Figure 2 of Traffic 18, 840, 2017, the accuracy in both particle tracking and localization of individual particles is affected by several factors such as SNR and particle density. All 15 tracking programs tested tend to become prone to mistracking when the particle number is increased by only a factor of 2 (500 to 1000 in 512^2 pixel images). More importantly, localization accuracy between truly paired particles (between ground truth and estimated localization of correctly identified particles) also becomes less accurate with increasing particle numbers in the simulated images. The latter may arise from two particles being close to each other with their signals partially overlapped. In addition, appearance and disappearance of particles due to fast F-actin turnover in some cell areas and binding kinetics of SiR-actin may increase the error. Because frequency and scale of errors associated with mistracking, misidentification of new tracks and inaccurate localization of clustered particles may vary and depend on the density of particles and turnover rates of the particles, the above assumption of Gaussian distribution is not acceptable as the method to estimate the distribution of tracking error.

These issues must also be considered when actin dynamics is compared between different cell structures such as actin stress fibers, focal adhesions and the cell cortex. Images in Fig. 2g contain fairly dense speckles in both live and fixed cells. Contrary to the statement, most of 'speckles' along stress fibers appear to consist of not a "single-molecule" but multiple SiR-actin molecules. The particles seem far more condensed than in the simulated particle images generated in the above Nature Methods paper. Uncertainty remains as to how much the estimated tracks differ from the true tracks in the analysis of such densely labeled particle images. This question would be extremely difficult to reasonably solve with any currently available technologies. It would also be impractical to test any phenomenological models using the data which might have been affected by various degrees of errors depending on the conditions and the categories.

Due to the lack of careful handling of problems and errors in particle tracking analysis, I can't be positive for publication of this paper.

Reviewer #3:

Remarks to the Author:

The manuscript by Miller, Korkmazhan, and Dunn details the development of a new analysis method of

actin Quantitative Speckle Microscopy (qFSM). They use a new, Bayesian statistical approach to estimate the velocity distributions of actin filament fiducials. This approach provide more accurate measurements of slow velocities, such as those presumably in the cell cortex of the cell body. This is an interesting advance that solves a noted problem with existing approaches. The manuscript is appropriately written for a specialized audience, but would still benefit from further clarification of its claims about past literature and assumptions within the new model.

1) The authors claim that the “87majority of F-actin tracking measurements, such as those supporting the molecular clutch model, use fluorescent fiducial densities that are too high to distinguish the motion of individual filaments (31).” It is unclear what, exactly, these authors are referring to in their claim that fiducial densities were too high in past work and that individual filaments cannot be or have not been tracked. The reviewers’ understanding of the Danuser software is that indeed, fiducial markers (labeled actin) within individual filaments are tracked. Afterwards, these flow tracks are correlated at a user-specified length. Flow is interpolated in time and averaged in space and time, at intervals controlled by the user. If the authors want to make this strong claim, sufficient detail about which step is problematic and how they know the fiducial densities and whether they’re too high should be explained. Otherwise, this claim could be removed without diminishing the need to an approach to deal with the noise in slow velocity tracks.

2) The authors claim that “a suitable measurement of single-filament dynamics in living cells has not been reported” is not helpful or clear. There is a significant body of literature in this field. What, specifically, is unsuitable in all these past works?

3) The authors claim that discontinuous or stick-slip flow is not incorporated in existing models estimating flow. The Molecular clutch model, including that put forth by Waterman et al., assumes that flow is constant/ fixed in simulations. Experimental data have shown that flow slows down at adhesions. Thus, it is implied in the existing molecular clutch models that actin flow increases during frictional slippage in which actin slips from the adhesion transmitting force. The authors should modify their representation of the existing models. <https://www.ncbi.nlm.nih.gov/pmc/articles/PMC6792288/>

4) The actin cortex under the cell body is assumed to be “anything within the cell mask, but not in the adhesions and actin stress fibers, defined as the brightest 2% of pixels.” This might be false assumption. How the limitations of the masking assumptions might affect the end conclusions should be discussed.

We thank the Reviews for their many helpful comments. As detailed below, we believe we have addressed all of them in full. As a result, the manuscript is much improved. Reviewer comments are provided *verbatim*. Our responses are provided in blue text.

Reviewer #1 (Remarks to the Author):

This is a very interesting and clearly written study that investigates small and slow displacements of elements of actin network in various segments of the ventral actin network of an adherent cell - in the cortex, stress fibers, focal adhesions. The authors focus on very small displacements, the problem with which is that measurement error overwhelms true displacements. To overcome this obstacle, the authors use an elegant Bayesian method, which can predict probability distribution for true displacements, based on distribution for errors that is measured for fixed actin network, but this prediction depends on an assumed functional form of the true displacements probability distribution. The finding is that the so-called Weibull distribution describes the data well. Furthermore, the authors show that the best model that agrees with these data is discrete stochastic jumps of a few tens of nm size, every few seconds.

Potentially, these are formidable results. Here are some critical comments (some of them probably stem from my ignorance or misunderstanding):

We are gratified by the Reviewer's overall enthusiasm for our manuscript. We sincerely thank the Reviewer for his/her critiques and suggestions; addressing them has considerably strengthened the study.

1) Relevance for cell biology: the results are very curious, but I am struggling to understand what is the impact of this knowledge. OK, so there are small and slow discrete displacements of actin filaments... Does it affect cell movement? Actin network turnover? Perhaps, there are implications for actin network rheology? Other than very vague discussion that the results imply elastic properties of the network (which I don't find convincing anyway), I don't see anything else... The authors might want to get together with someone like Fred MacKintosh to think further.

We have revised the Discussion section of our study in response to this comment. To summarize, our results support a model in which motion in the actin cytoskeleton is best understood in terms of short-lived equilibria punctuated by transient jumps in position. In a major improvement relative to the previous version of the manuscript, we now present a single, unified model that can account for F-actin dynamics across all regions of the cytoskeleton we examined, and across a broad range of timescales (see point 5 below). This advance was directly due to the Reviewer's critique (thanks!). The implication is that one physical description of F-actin motion may hold across a broad range of cytoskeletal structures despite their apparent differences in organization and cellular function. To us this is an exciting possibility, as it suggests that the emergent properties of the actin cytoskeleton may be relatively understandable despite its complexity at the molecular scale. We've attempted to convey this point (with appropriate caveats) in the Discussion:

“We used numerical simulations to fit the same data to a mechanistic model of actin motion. Surprisingly, the discontinuous jump model fit the data significantly better than the more common, continuous diffusion with drift framework for all populations, despite the diversity of organization and function of these structures. While the analytically tractable 1D position jump is sufficient to fit our measured displacement distributions for a given timescale, we found that allowing directional reversals is sufficient to recapitulate our data across timescales with only 3 fit parameters. The ability to describe a variety of F-actin populations with a relatively simple model suggests that the emergent physical properties of the cytoskeleton may be understandable within a common physical framework despite the considerable complexity of the cytoskeleton at the molecular scale.”

2) There is a good discussion that the inferred displacements could be the result of crosslinking- and/or myosin-pulling events. The obvious suggestion: the study would be much more convincing, if the authors do relatively simple experiments: 1) use blebbistatin and caliculin to perturb contractility; 2) inhibit and overexpress alpha-actinin or other crosslinkers; 3) use latrunculin and jasplakinolide to change actin network. By the way, the characteristic displacements probably have nothing to do with helical structure of actin filaments, but very likely are the same order of magnitude as the actin mesh size - this also could be tested.

These are great suggestions. As expected, when we treated the cells with blebbistatin, we observed an overall loss of stress fibers and large adhesions. In addition, we found a concentration of Latrunculin A that caused cell morphology to become more irregular (many long, stretched cells and cells with long irregular protrusions) compared to a matched DMSO control, while still preserving sufficient cytoskeletal integrity to provide trackable puncta. These data are now presented in Figure S10. Latrunculin A treatment resulted in increases in both average jump distance and reversal probability in both stress fibers and the cortex, perhaps due to the loss of structural organization (e.g. larger mesh size, lack of oriented forces) due to a thinning of the actin meshwork. In contrast, we did not observe notable changes in either the raw data or fit parameters for cells treated with blebbistatin. This result may reflect the experimental choice to use concentrations that preserved at least a few stress fibers and adhesions, such that cells could be more directly compared to DMSO controls. Alternatively, it is possible that while blebbistatin altered the presence of stress fibers, the structures that remained were able to maintain similar dynamics to those in control cells:

In our hands Calyculin is a tricky reagent: too little and the cells don't respond, too much and they quickly detach from the substrate. The difficulty of getting reproducible results with this reagent made us pessimistic that Calyculin treatment would yield trustable data. Jasplakinolide is a good idea in principle, but unfortunately SiR-actin binds to F-actin via a jasplakinolide moiety, and when we added jasplakinolide it competed SiR-actin off F-actin, making tracking impossible. Finally, while it would be very interesting to tune the number of crosslinkers, this proved to be technically difficult. When knocking down alpha-actinin (1 and 4) with siRNA in HFFs, we saw a significant (~90%) reduction in mRNA by RT-PCR but did not see a similar

reduction in expression as evaluated by immunofluorescence, indicating that these cells maintain tight control over alpha-actinin levels at the protein level.

Importantly, this comment prompted us to perform the actin tracking measurement in untreated cells at both 37 °C and 26 °C (Fig. S9). We observed a noticeable decrease in actin displacements at the lower temperature. Upon fitting to the jump model (see point 5, below), we find that jump distances are essentially unchanged, while the waiting time between jumps increases by roughly a factor of 2 at 26 °C (Figure S9), in line with typical temperature sensitivities of biological macromolecules. These new data provide strong supporting evidence for the jump model.

“We next examined how the motion of individual actin filaments varied in response to perturbations known to affect cytoskeletal dynamics at a more global level. The kinetics of biological macromolecules are temperature sensitive, with rates that typically change by a factor of 2-4 with every 10 °C change in temperature (Rossi et al 2005, Yengo et al 2012). When we quantified F-actin dynamics in HFFs imaged at a lower temperature (26 °C), we found that F-actin motion was best fit by similar jump distances, but less frequent jumps when compared to measurements at 37 °C (Figure S9). We also explored the effect of decreasing the overall abundance of F-actin by treating cells with latrunculin A, which depolymerizes F-actin and sequesters G-actin monomers (Fujiwara et al 2018). F-actin velocities in HFFs treated with 200 nM latrunculin A were best fit by increased jump distances and waiting times between jumps, as well as an increased probability of reversal, suggestive of a disruption of the cytoskeletal network (Figure S10). The effects of treatment with 25 μM blebbistatin, a concentration chosen to preserve the presence of some stress fibers and adhesions, were more muted (Figure S10).”

Finally, thank you for the insightful comment on the actin mesh size—the actin mesh size is indeed consistent with our best fit jump distances, and we have added the following to the discussion of the paper:

“In this regard it is intriguing that the magnitude of jump distances ~50 nm is similar to the actin network mesh size (10s-100s of nanometers) (Morone et al. 2006).”

3) Why are the authors sure that what they observe is displacements of individual filaments, and not more complex network structures?

This point was unclear in the original text. We label at concentrations with which we can ascertain motion of single fluorophores (which are linked to single filaments), but the motion that we measure may indeed be a combination of single filament motion relative to local complex structures and mass motion of those complex structures. We have altered the wording of the paper to read:

“We note that, while these measurements report on the motion of individual filaments, these motions are likely composed of a combination of individual filament motion and the coordinated movements of locally crosslinked structures.”

4) I could not figure out from the text: are the true displacements always isotropic, and their directions uncorrelated between consecutive movements? For the cortex, this is very possible, but I find this very puzzling for stress fibers, where there is constant slow flow along the fiber's length, and for adhesions, where the displacements are supposed to be biased in the direction of the center of the fiber on which end the adhesion is. Again, some myosin perturbations may give a clue here.

In the revised text, we find that a one-dimensional jump model with reversals adequately describes F-actin displacements (see point 5, below). That this model fits all four F-actin populations examined is somewhat unexpected. Several additions to the model (2D motion, velocity instead of position jumps) may more closely represent reality, but we find that this relatively simple model is sufficient to fully capture the data we've measured. This is discussed further in response to point 5 below.

We also note that the average timescale of our tracks is <10 s and the longest timescale we fit is 40 s. For the timescales investigated here, it is possible that for a majority of tracks, the motion maintains sufficient directionality or is sufficiently confined to 1D motion for the jump model to adequately capture the data.

5) I find it extremely disturbing that the characteristic time between jumps grows with observation time. Clearly, the authors are also bothered by this, but the guess in the discussion that this has something to do with a more stable actin population does not sound convincing to me. Speaking of which: the actin network is supposed to turn over on time scales similar to measurement times. First of all, the turnover time probably could be estimated by FRAP, or some other techniques. Second, were there discarded shorter trajectories? Basically, the issue of the turnover should be more carefully discussed. Relevant to that - there are likely some displacements away from the focal plane, no?

The timescale-dependence described in the earlier version of the manuscript bothered us too. In revising the manuscript, we considered many alternative models that could potentially describe F-actin dynamics across timescales. These included velocity-jump models (where the filament makes abrupt switches between velocities), and 2D models in which the filament was allowed to turn, analogous to run-and-tumble dynamics. In the end, what turned out to describe the data best was a 1D position jump model (as in the earlier version) but with a finite probability of direction reversal at each jump. This model was not analytically tractable, at least for us. However, we could derive the fit parameters using an iterative approach in which a probability density function (PDF) for a given model was generated via Monte Carlo simulation, and then fit to the actual data using an MLE framework. This model works well: it captures F-actin displacements across timescales (Fig. 4e,g), with physically reasonable differences in fit parameters for stress fibers vs. cortical actin. As discussed above, it also yields sensible fit parameters for perturbations such as drug treatments and alterations in temperature. We thus feel that this additional work has considerably enhanced the study.

With regard to turnover time, this can in principle be estimated by comparing the distribution of puncta lifetimes measured for fixed cells (where F-actin cannot turn over) vs. live cells. The puncta lifetimes in both live and fixed cells are well-fit by a biexponential distribution, with rates in the fixed cell population of $r_1 = 0.06/\text{s}$ and $r_2 = 0.33/\text{s}$. These rates likely represent turnover-independent processes like bleaching and Sir-Actin unbinding. The presence of two distinct lifetimes is not what one would expect for typical reversible binding. However, previous work indicates that jasplakinolide (the actin binding moiety in SiR-actin) binds to F-actin in at least two modes that may depend on F-actin nucleotide state (Spector et al. 2000).

In live cells, the rate of F-actin turnover is expected to add to these rates, as the observed rate of SiR-actin disappearance should reflect the combined rates of bleaching, Sir-Actin unbinding, and F-actin turnover, whether through depolymerization or movement out of the TIRF excitation field. Comparison of the live- and fixed-cell data allowed us to extract approximate F-actin turnover rates of $0.06/\text{s}$ for cortical F-actin and $0.03/\text{s}$ for stress fiber F-actin, giving lifetimes of 18 and 34 s, respectively. These timescales are compatible with previous FRAP measurements (Campbell and Knight 2007, Fritzsche et al 2013, Smith et al 2013, Skamrahl et al 2019). These results are described in Supplementary Note 4 and Figure S11.

To address the second point, shorter trajectories are not discarded, but for each trajectory, displacements are calculated across all possible timescales from the acquisition timescale up to the total timescale of the track or 40 s, whichever is shorter. As such, longer tracks are included in both the short and long timescale populations, while short tracks are only included in the short timescale population.

It is true that there are likely displacements away from the focal plane. This may contribute to some inaccuracy in our estimated jump displacements since we do not have precise z-localizations to allow us to make these measurements in 3D. However, we can rule out dramatic motion in z as particles that move out of the focal plane likely either 1) move out of the TIRF plane or 2) fail localization and are no longer tracked.

Reviewer #2 (Remarks to the Author):

The authors attempt to infer the hidden characteristics in the motion of actin filaments in live cells. To do so, they developed a method to statistically subtract the tracking error distribution from the observed distribution of F-actin speckle displacement in time series images. Although the scope of this paper may potentially be important to the research field, the paper lacks careful consideration of the mechanisms of error generation in object tracking for low signal to noise ratio (SNR) data.

The method developed in this paper uses a very simplified assumption in estimating the distribution of tracking errors; that is “measured coordinates are Gaussian distributed around the true location ...” in Supplementary Information p2. This is overwhelmingly too simplified. As

found in Figure 2 of Nature Methods 11, 281, 2014 and Figure 2 of Traffic 18, 840, 2017, the accuracy in both particle tracking and localization of individual particles is affected by several factors such as SNR and particle density. All 15 tracking programs tested tend to become prone to mistracking when the particle number is increased by only a factor of 2 (500 to 1000 in 512^2 pixel images). More importantly, localization accuracy between truly paired particles (between ground truth and estimated localization of correctly identified particles) also becomes less accurate with increasing particle numbers in the simulated images. The latter may arise from two particles being close to each other with their signals partially overlapped. In addition, appearance and disappearance of particles due to fast F-actin turnover in some cell areas and binding kinetics of SiR-actin may increase the error. Because frequency and scale of errors associated with mistracking, misidentification of new tracks and inaccurate localization of clustered particles may vary and depend on the density of particles and turnover rates of the particles, the above assumption of Gaussian distribution is not acceptable as the method to estimate the distribution of tracking error.

These issues must also be considered when actin dynamics is compared between different cell structures such as actin stress fibers, focal adhesions and the cell cortex. Images in Fig. 2g contain fairly dense speckles in both live and fixed cells. Contrary to the statement, most of 'speckles' along stress fibers appear to consist of not a "single-molecule" but multiple SiR-actin molecules. The particles seem far more condensed than in the simulated particle images generated in the above Nature Methods paper. Uncertainty remains as to how much the estimated tracks differ from the true tracks in the analysis of such densely labeled particle images. This question would be extremely difficult to reasonably solve with any currently available technologies. It would also be impractical to test any phenomenological models using the data which might have been affected by various degrees of errors depending on the conditions and the categories.

Due to the lack of careful handling of problems and errors in particle tracking analysis, I can't be positive for publication of this paper.

We sincerely thank the Reviewer for their time in evaluating our manuscript. We realized upon re-reading the original text that we failed to include relevant details about how we select puncta that can be reliably tracked. The addition of this information to the manuscript, along with a few improvements we have made to our analysis pipeline, are detailed below.

For the Reviewers' convenience, we first present an **Overview** of our tracking methodology. We then detail **Incremental Improvements** we made in the process of preparing this revision. Finally, we describe **Tests for Robustness** that we performed to assure that any imperfections in our methodology did not alter the conclusions of our study. All of these details are included in the revised manuscript and supplemental information.

1) Overview

We agree that it is challenging to track dense fields of single molecules. Instead, as explained below, we sought to confine our analysis to the subset of puncta within a field of view that are definitively trackable. This is much easier in a technical sense than trying to track every punctum.

We found in our initial optimization of our analysis pipeline that common tracking software packages are prone to mistracking in the datasets tested. These errors were obvious by eye as aberrantly large displacements or tracks that followed one puncta and then jumped to another. We found QFSM (Mendoza et al 2012) to perform best at tracking large numbers of puncta while avoiding making the false linkages we noticed in other software packages. The criteria within QFSM were set to be quite strict, such that only a fraction of all visible puncta are tracked, corresponding to those that one can easily follow by eye. In consequence, the densest areas are often not tracked (see Supplementary Video S2 for an illustration of this point). This feature allows us to avoid attempting to track puncta that are too close together. Importantly, photobleaching during the course of data acquisition yields trackable puncta within stress fibers and other dense regions in the latter half of a typical movie. This strategy allowed us to retrieve sufficient tracks from both relatively sparse and dense regions of the cytoskeleton. Tests for the effects of fluorophore density are detailed in Tests for Robustness, below.

Stills from the first (left) and last (right) frames of Video S2, where circles show localizations from QFSM. Note that very few localizations are not made in the densely labeled stress fiber region in the first frame. Magenta circles are puncta which are discarded by our subpixel localization (see below), while green circles are puncta which are successfully localized. Video was denoised using noise2void.

On top of the selective tracking by QFSM, which gives pixel-level localizations, we layer our own subpixel localization. Specifically, we fit each track localization to a Gaussian in x and y , and discard any localizations that fail this fitting: failure to fit includes the best-fit center being ≥ 2 pixels (128 nm) away from the original, pixel-level localization from QFSM, or the fit Gaussian having an unrealistically low or high fit standard deviation, σ . For a typical example dataset, unaccepted center positions account for $\sim 7.3\%$ of all localizations, while unaccepted standard deviations account for $\sim 40.8\%$ of all localizations. There is high overlap between these two groups, such that together these include $\sim 40.8\%$ of localizations (only 0.02% of localizations have an unaccepted center but an accepted standard deviation). Overlapping particles are likely

to have poorly fit centers or high σ 's, and so are avoided using these metrics. We will discuss this point further below (see *Effect of allowable localizations on fit quality*).

These selection criteria are now included in *Materials and Methods, Actin Tracking Analysis*.

2) Incremental improvements to tracking made during the revision

In the course of revising the manuscript we made some additional improvements that enhance the stringency of our data analysis:

1) *Testing the effect of allowable localizations on fit quality*: In theory, the best fit Gaussian to a single molecule should have a σ of $\sim 0.25 \cdot \lambda / \text{NA} = 113$ nm, where λ , the wavelength of light, is ~ 675 nm, and NA for our objective is 1.49. However, we find during Gaussian fitting to puncta that the average best fit σ is ~ 260 nm, with the larger value plausibly due to the low signal-to-noise for single molecules. In refining our methodology, we first tried tightened the stringency for acceptance with respect to σ to a range of 75-170 nm, or within a factor of 1.5 from the theoretical $\sim 0.25 \cdot \lambda / \text{NA}$. We found this tighter range improved the fit of fixed cell data to the noncentral chi distribution relative to a more permissible range of σ values (30-600 nm). In fact, the smaller the acceptable range for σ is made, the more closely the displacement distribution matches a noncentral chi distribution. This observation supports the assumption that Gaussian-distributed localization errors are an accurate descriptor for our measured localizations (see also point 3 below).

2) *Denoising*: A cost to the use of a stringent value range for σ was that we lose a large number of real localizations. To differentiate between large σ 's due to noise (which we would like to keep) and those due to overlapping particles (which we want to exclude), we applied a deep-learning based denoising technique, noise2void (Krull et al, arXiv 2019), to our data. After treatment with noise2void, best-fit σ 's shift to smaller values (average ~ 150 nm from ~ 260 nm) and we are able to employ a strict limit on fit σ 's (using the theoretical value above +/- a factor of 1.5) while also retaining a large fraction of our localizations ($\sim 50\%$).

3) *Accounting for variation in localization error*: It is reasonable to suppose that the localization error for a given fluorophore will vary stochastically, and that this variation can account for the range in σ values discussed in point 1, above. As an independent check, we assessed the variability in fit position for our long tracks (20+ localizations) in fixed cells. Here, the standard deviation for x and y positions should be equivalent to the σ_{xy} in our noncentral chi fit. As might be anticipated, we find that for these tracks there is indeed a range of localization errors, which is why the noncentral chi with a single σ_{xy} did not perfectly fit the fixed cell data. However, this range of localization errors is adequately captured by a mixture of two chi distributions, each with a different sigma, σ_{xy1} and σ_{xy2} . We have updated our fits to use this mixture model in all of our calculations.

Please note that although the inclusion of (2) and (3) above improved the quality of our data fitting, this did not alter the qualitative features of the data or their interpretation.

Test for robustness

In order to test the effects of various types of tracking errors on our conclusions, we selectively reanalyzed subsets of our data:

1) *Punctum brightness*: we sorted tracks into three groups based on their intensity (from the amplitude of the Gaussian fit during subpixel localization). These three groups were the dimmest 25% of tracks, the brightest 25% of tracks, and the middle 50% of tracks from each population. We find that the dimmest tracks yield the highest measured displacements (likely because they have the greatest localization error), and make up only a small fraction of long tracks (again likely due to poor localization of the tracks causing failed tracking). However, when we fit each of these populations to a Weibull distribution, we find that we get similar results. Though the results from the dimmest tracks are noisy, and give slightly larger scale values, all three populations show similar fit values (shapes near 1-1.5 and scales which vary similarly with timescale). These data are included in SI, and reproduced below:

Figure S13. Displacements of tracks with varying punctum brightness and Weibull fits to the same.
a) Measured displacements at three timescales (2 s, left; 10 s, center; and 20 s, right) for the dimmest 25% (blue), the brightest 25% (yellow), and the remaining 50% (orange) of tracked puncta.
b) Best-fit Weibull parameters to the same populations, across timescales from 2 s to 36 s. These data represent the cortical population from one experimental replicate.

2) *Track length*: we similarly tested the Weibull fits to displacements pulled from tracks of varying length, at three timescales: 4 s, 8 s, 16 s. For each timescale, we can calculate displacements for tracks of that length or longer (See Figure S14, reproduced below). Across fitable timescales, we get close matching of shape parameters (1-1.4) but find that the scale parameter varies by approximately a factor of 2 with increasing track lifetime for a given observation timescale. This may be due to differences in the underlying F-actin (e.g. longer lasting actin filaments move more slowly), sampling (e.g. slower moving filaments are more easily and more often tracked) or localization error (e.g. dimmer fluorophores with higher localization error are interpreted as larger displacements but are not able to be tracked as long). While we didn't attempt to definitively distinguish among these possibilities, we did test the effect of these differences on our final model by fitting stress fiber and cortical populations from one day's dataset to the 1D jump model with reversals, either with all tracks included, or with only tracks of ≥ 16 s length (where scale parameters start to level off, see below). We see modest differences in jump times, jump distances, and the probability of reversal. The largest difference was for the reversal probability in stress fibers. However, this parameter tends to be the most poorly constrained in the model. In any case, these variations, if real, do not alter the principal mechanistic conclusion of our study, namely that a jump model fits F-actin motion better than diffusive models.

Figure S14. Best-fit Weibull parameters for tracks of varying durations, measured at three time intervals: 4 s (lightest blue), 8 s, and 16 s (darkest blue). These data are drawn from the cortical population from one experimental replicate.

Table S2: 1D distance jump with reversals model fit to stress fiber and cortical data from one experimental replicate.

	stress fibers, all tracks	stress fibers, tracks > 14 s	cortical, all tracks	cortical, tracks > 14 s
average time between jumps (s)	6.9	8.9	3.6	5.6
average jump distance (nm)	54.5	49.4	61.5	61.3
reversal probability	0.26	0.03	0.37	0.19

3. *Tracking at varying labeling densities:* As noted by the reviewer, we label at relatively high densities. This allows us to identify stress fiber locations in the cell. However, there is significant bleaching of the sample over the acquisition time. We find that the mean intensity of the image drops by ~50% from the first frame to the tenth frame. As noted above, QFSM tends to avoid tracking in the dense stress fiber regions during early frames, and tracks in this region often come from later frames, when the region has bleached significantly. As a test of the effect of speckle density on tracked displacements, we compared the speckle displacements over 10 seconds of speckles from the first twenty frames vs. those from the last twenty frames of a 60 frame movie. In this case, we observe similar distributions (shown below), despite the difference in overall density.

Fig S12. Measured displacements in live cells (all subcellular populations) over a 10 s time interval, measured either during the first 20 s (10 frames) of a movie, or from the last 20 s of the movie.

It is also worth noting that the HUVEC samples were obtained at lower speckle densities than those in HFFs. Despite this difference we obtain qualitatively similar fit parameters to the jump model in this cell line (SI Fig S8). Finally, localization errors for puncta that met these quality control metrics for SiR-Actin-labeled fixed cells were similar over a 100-fold range in probe densities.

Best fit sigma values from the noncentral chi distribution fits to displacements measured in fixed cells. Cells were labeled with varying SiR-Actin dilutions, both above and below those used in our measurements (1 in 50k dilution). Values are fit for varying timescales, from 2 to 32 s.

This evidence gives us confidence that our selection criteria (above) are successful and our measurements are not noticeably influenced by fluorophore density.

Reviewer #3 (Remarks to the Author):

The manuscript by Miller, Korkmazhan, and Dunn details the development of a new analysis method of actin Quantitative Speckle Microscopy (qFSM). They use a new, Bayesian statistical approach to estimate the velocity distributions of actin filament fiducials. This approach provide more accurate measurements of slow velocities, such as those presumably in the cell cortex of the cell body. This is an interesting advance that solves a noted problem with existing approaches. The manuscript is appropriately written for a specialized audience, but would still benefit from further clarification of its claims about past literature and assumptions within the new model.

We are gratified by the Reviewer's overall positive evaluation, and have endeavored to comprehensively address the identified areas for improvement.

1) The authors claim that the "...majority of F-actin tracking measurements, such as those supporting the molecular clutch model, use fluorescent fiducial densities that are too high to distinguish the motion of individual filaments (31)." It is unclear what, exactly, these authors are referring to in their claim that fiducial densities were too high in past work and that individual filaments cannot be or have not been tracked. The reviewers' understanding of the Danuser software is that indeed, fiducial markers (labeled actin) within individual filaments are tracked. Afterwards, these flow tracks are correlated at a user-specified length. Flow is the interpolated in time and averaged in space and time, at intervals controlled by the user. If the authors want to make this strong claim, sufficient detail about which step is problematic and how they know the fiducial densities and whether they're too high should be explained. Otherwise, this claim could be removed without diminishing the need to an approach to deal with the noise in slow velocity tracks.

We apologize for the confusion on this point. In the cited review, the distinction is drawn between tracking single molecules and tracking speckles comprised of multiple fluorophores (and the point is made that multiple fluorophore tracking is often done because the speckles are brighter, less likely to bleach, and thus easier to track).

We equate single fluorophore tracking with single filament tracking. While it is true that multiple fluorophores can be tethered to a single filament, due to the small width of single filaments and the high density of filaments in many of these structures, it's unlikely that multiple fluorophore speckles are always representative of single filaments. We have reworded this to be clearer:

"Similarly, the large majority of F-actin tracking measurements, such as those supporting the molecular clutch model, employ speckle microscopy, in which F-actin is labeled at a density such that puncta typically comprising several fluorophores are tracked. This technique, though powerful, does not straightforwardly report on the motion of individual fluorophores, and hence individual filaments."

2) The authors claim that "a suitable measurement of single-filament dynamics in living cells has not been reported" is not helpful or clear. There is a significant body of literature in this field. What, specifically, is unsuitable in all these past works?

Our wording was inexact. We have changed this passage to read: "However, to our knowledge no measurement of single-filament dynamics in living cells has been reported that differentiates between continuous flow and discontinuous slip-stick models of motion."

3) The authors claim that discontinuous or stick-slip flow is not incorporated in existing models estimating flow. The Molecular clutch model, including that put forth by Waterman et al., assumes that flow is constant/ fixed in simulations. Experimental data have shown that flow slows down at adhesions. Thus, it is implied in the existing molecular clutch models that actin

flow increases during frictional slippage in which actin slips from the adhesion transmitting force. The authors should modify their representation of the existing models.

<https://www.ncbi.nlm.nih.gov/pmc/articles/PMC6792288/>

We agree with the Reviewer's description of previous studies. What is new here, relative to canonical molecular clutch models (for example Chan and Odde, Science 2008) is that these earlier models treat the actin cytoskeleton as a crosslinked monolith in which all filaments move in synchrony. Our results in this study are consistent with a revised model in which individual filaments are allowed to move independently (Tan et al. Sci Adv. 2020). In this revised model, filaments exhibit long periods of stasis, punctuated by movement that is jump-like on the measurement timescale. We now clarify this issue:

“Recent results from our laboratory implied that actin filaments attached to integrin-based adhesions do not undergo continuous retrograde flow, but instead move with discontinuous stick-slip motion (Tan et al. 2020). This picture contrasts with earlier molecular clutch models, which featured continuous, retrograde F-actin flow (Chan and Odde 2008). Discontinuous movement at the level of individual filaments would be consistent with a description of the cytoskeleton as a gel or glass (Zaccarelli 2007, Mandadapu et al 2008). However, to our knowledge no measurement of single-filament dynamics in living cells has been reported that differentiates between continuous flow and discontinuous slip-stick models of motion.”

4) The actin cortex under the cell body is assumed to be “anything within the cell mask, but not in the adhesions and actin stress fibers, defined as the brightest 2% of pixels.” This might be false assumption. How the limitations of the masking assumptions might affect the end conclusions should be discussed.

To clarify, actin stress fibers are masked from the brightest 2% of pixels from a time projection of the actin video. Because the stress fibers are dense with actin and stable over our observation time, we tend to see multiple fluorophores binding in the same location over the course of a recording in these regions. Still, the stress fibers are somewhat difficult to capture due to incomplete labeling, and so a mask is chosen to balance including as much of the stress fibers as possible while excluding as much of the cortex as possible.

In practice, we find a cutoff of 2-3% to best separate these two populations by eye, and find that these values also lead to good separation of behavior (i.e. displacement distributions). At stricter cutoffs (like 0.1%), too many stress fiber puncta are included in the cortical population and both populations become more similar. Likewise, at looser cutoffs (like 5%), too many cortical puncta are included in the stress fiber population and both populations again converge.

Reviewers' Comments:

Reviewer #1:

Remarks to the Author:

I commend the authors for considerable efforts and find the manuscript much improved and ready for acceptance

Reviewer #2:

Remarks to the Author:

My previous review focused only on the validity of the noise elimination method. This is because the data in Fig. S1 showed where the problem is. If the particle tracking is completely error-free, the parameters should remain the same over time. In current Fig. S1, however, σ_1 and σ_2 increase gradually to a large extent (by >50%). This suggests that tracking error still remains to be improved. In addition, the σ values range from 30 nm to ~90 nm, which are not so good in the current technological standard. I am not convinced that the data are sufficiently accurate to infer any of the nanometer-scale mechanisms.

More critically, I have serious concerns over the use of 'F-actin velocity distribution' in the model comparison. F-actin, especially in stress fibers, generally shows constant velocity, conveyed by the local network flow and internal contractile motions. To analyze such motion, it is mandatory to retain the information about individual motions along the time. On the contrary, such crucial information is discarded in 'F-actin velocity distribution'. This is like trying to infer how road cars change the speed in a city after breaking the speed sequence of individual cars. Many unrealistic models would conveniently explain such broken speed distribution. I am also concerned with the lack of consideration of the heterogeneity and the physical properties of F-actin inside cells, in both distribution and model analyses.

Reviewer #3:

Remarks to the Author:

The manuscript is much improved in terms of accessibility, advance in understanding the movement of actin, and methodology. The new findings on the jump model explaining the behavior of 4 difference actin structures and the clarified methods and model assumptions make this an interesting work worthy of publication.

The remaining concern is on significance. Statistical analysis on the differences between actin structures presented in Figure 4g should be done in order to determine whether the differences have the potential to be biologically significant. Furthermore, the reasoning of how the computed fit times, distances, and reversal probabilities for each actin structure might explain the cellular behavior of those structures should be explicitly described. The discussion speculates that these are at play, but assumes that the reader has the knowledge on hand to connect this speculation to each actin structure. Please add the rationale behind each parameter difference for each relevant actin structure to the discussion.

Reviewer #1 (Remarks to the Author):

I commend the authors for considerable efforts and find the manuscript much improved and ready for acceptance

We are gratified by the Reviewer's response! Thank you for your time in considering our work, and for the comments that have resulted in material improvements to our study.

Reviewer #2 (Remarks to the Author):

My previous review focused only on the validity of the noise elimination method. This is because the data in Fig. S1 showed where the problem is. If the particle tracking is completely error-free, the parameters should remain the same over time. In current Fig. S1, however, sigma 1 and sigma 2 increase gradually to a large extent (by >50%). This suggests that tracking error still remains to be improved. In addition, the sigma values range from 30 nm to ~90 nm, which are not so good in the current technological standard. I am not convinced that the data are sufficiently accurate to infer any of the nanometer-scale mechanisms.

For context, Figure S1 showed data that we used to estimate the distribution of localization errors for control, fixed cells. Although we did not make it clear in the previous draft, the increase in sigma reflects gradual sample or focal drift that remained even after drift correction was applied. This is illustrated in the modified Figure S1 (below), which shows that in fixed cells there is a slow increase in distance over time between initial and subsequent localizations for the same punctum (averaged across all long-lived (≥ 40 s) puncta for 9 fixed cells). This accounts for the gradual increase in standard deviation (σ) with time, but does not actually affect our localization accuracy, which is determined based on the shortest time interval (2 s), for which drift is negligible. This misunderstanding was very reasonable given the way the data were portrayed in the previous draft.

To address this, we now present a plot of residual drift in Figure S1B. We have also modified Figure S1 to remove the confusing non-central chi fits for time delays longer than 2 s, as the same information is better conveyed by the direct measurement of residual drift in the new supplemental figure.

Fig. S1. A. Displacement distributions in fixed cells measured over the 2, 10, 20, 30, and 40 s timescales. B. The net displacement vs. time, averaged across all long tracks (≥ 20 s) in $n=9$ fixed cells (solid), and a best linear fit (dashed). The shaded region represents the timescale regime in which our live cell measurements are made. The displacement is normalized by subtracting the distance after 1 frame, which reflects only localization error. The steady increase in net distance is consistent with residual drift.

Importantly, the amount of residual drift at our longest experimental timescale (40 s) for live-cell measurements is ~ 15 nm, which is small relative to the average distances puncta moved on this timescale. Thus, sample drift does not materially affect our results.

These points are now clarified in the main text:

“Over varying timescales of analysis (2 - 40 s) using constant, 0.3 s exposures, the distribution of measured distances in fixed cells was similar, but with a small enrichment of longer displacements over longer timescales (Supplementary Fig. 1a). This effect likely stems from residual sample or focus drift evident in the tracks (Supplementary Fig. 1b), amounting to only ~ 20 nm at the 40 s timescale. This drift does not affect the localization error, which is determined based on the shortest time interval (2 s), where drift is negligible.”

Parenthetically, our localization accuracy is comparable to studies in which single fluorophores are localized in the context of living cells, for example Rossier *et al.*, NCB 2012 (47 nm); Ezedin *et al.*, eLife 2014 (70 nm); Sungkaworn *et al.*, Nature 2017 (26 nm); and Bosch *et al.*, Biophysical Journal 2014 (40 nm).

More critically, I have serious concerns over the use of ‘F-actin velocity distribution’ in the model comparison. F-actin, especially in stress fibers, generally shows constant velocity, conveyed by the local network flow and internal contractile motions. To analyze such motion, it is mandatory to retain the information about individual motions along the time. On the contrary, such crucial information is discarded in ‘F-actin velocity distribution’. This is like trying to infer how road cars change the speed in a city after breaking the speed sequence of individual cars.

Many unrealistic models would conveniently explain such broken speed distribution. I am also concerned with the lack of consideration of the heterogeneity and the physical properties of F-actin inside cells, in both distribution and model analyses.

We address these three related critiques separately:

First, as the Reviewer notes, movies of stress fiber F-actin at the ensemble level convey the impression of continuous movement. For example, confocal microscopy measurements reported by our group and others show that labeled F-actin or α -actinin can in some cases appear to flow continuously out of adhesion sites and into the stress fiber. In other cases F-actin or α -actinin tracked in this way appears to move in a bidirectional fashion, both toward and away from adhesion sites (Owen et al. MBoC 2017). In either case, it is important to note that typical measurements are multiple minutes long, a timescale at which individual F-actin filaments turn over multiple times. In short, so far as we are aware, the continuous movement to which the Reviewer refers applies at the ensemble, not single-filament, level.

In this study we do not attempt to describe the emergent properties of the stress fiber as a whole. Instead, we measure the motion of single molecules, and find these movements to be inconsistent with a model in which individual particles move at a constant velocity (Fig. 4). Our findings are however consistent with previous studies whose results implied heterogeneity in the velocities of individual actin filaments (Yamashiro et al. MBoC 2014), and in particular that a subset of F-actin filaments should have zero actin velocities (Tan et al. Science Advances 2020; Driscoll et al. PNAS 2020). We emphasize that the single-molecule and bulk measurements are not inconsistent: relatively rapid and heterogeneous movements at the single-filament level, averaged over many filaments and multiple filament lifetimes, are expected to yield the ensemble-level movements seen previously.

We have added to the response letter below a montage showing individual F-actin tracks from stress fibers. These data illustrate that a subset of puncta are essentially stationary, and that multiple puncta that show marked changes in effective velocity. Neither observation is consistent with a constant-velocity type model, but both are consistent with a jump-type model. As shown in Fig. 1d, *there is no obvious spatial pattern in this heterogeneity*. Thus, absent evidence to the contrary, we have stuck with the simplest assumption, which is that the movement of individual actin filaments is essentially statistical in nature (see below).

Examples of several long tracks from stress fibers.

With regard to the **second point**, we agree that individual F-actin tracks could in principle contain additional information about the underlying dynamics. However, manual analysis of individual tracks did not reveal obvious patterns of motion indicative of non-Markovian processes analogous to the automobile traffic patterns referenced by the Reviewer (see above). This is perhaps to be expected given that the timescale of our measurement, and of F-actin turnover, is smaller than the timescales of bulk F-actin flows in stress fibers and lamella.

We address this point in the following ways:

1) We note that the Rice model (Fig 4C), which models 2D diffusion plus constant drift, does not fit the data as well as a jump model.

2) We now more thoroughly discuss position vs. velocity jump models. The two models differ in that a position jump model assumes the particle is stationary between jumps, whereas the velocity jump model assumes that the particle undergoes periodic switches in velocity, and moves at constant speed between these switching events. Importantly, the outputs of the position and velocity jump models are, at a functional level, very similar for our study. Critically, both jump models allow particles to change their effective velocities within an individual track, which is necessary to fully account for our data (e.g. Fig. 4). We were agnostic between the two models, but focused on the position jump model because *i)* force measurements at individual integrins are consistent with position jumps (Tan *et al.* Science Adv. 2020), *ii)* the movement of temporarily unrestrained F-actin would be expected to be jump-like on our measurement timescale, and *iii)* the position jump model without reversals is analytically tractable. (A countervailing consideration is that a velocity-jump model would recover continuous movement, for example due to local viscous relaxation.)

It is important to emphasize that we do not claim that the statistical model we present explains the motion of all actin filaments under all circumstances, a point that we emphasize in the revised manuscript (for example the quote below). However, for a three parameter model it does a remarkably good job of describing F-actin motion in a variety of cellular contexts, and in the presence of a variety of perturbations. One likely reason for this concordance is that a jump model effectively captures heterogeneity in F-actin motion, for example a mixture of stationary and moving filaments, that is missing in other physics-based models. The current draft addresses these points explicitly as below:

“It is plausible that F-actin motion in the lamellipodia is dominated by advective flow, i.e. drift, with respect to the laboratory rest frame, as in previous studies that focused on lamellipodial F-actin dynamics. Whether filaments additionally undergo jump-type motion within flowing lamellipodial/lamellar actin remains to be determined.”

and

“It is also possible that while the position jump model is a useful approximation of the F-actin motion in this system, alternative models may more accurately reflect the underlying motion. For example, a velocity jump model, in which particles move at constant velocity before "jumping" to a new velocity, is functionally similar to the position jump model at the spatiotemporal resolution of our measurement. Both of these models inherently capture local heterogeneity of F-actin velocities in both space and time, consistent with previous single molecule observations of F-actin motion (Yamashiro *et al.* 2014, MBoC; Tan *et al.* 2020, Science Advances).”

To our understanding, a velocity jump model is consistent with the picture the Reviewer proposes. We feel that rewriting the manuscript to compare position- and velocity-jump models more thoroughly has strengthened the study, and thank the Reviewer for

prompting us to clarify this point. We have additionally modified Fig. 4 to make the close correspondence of the position and velocity jump models clear.

Third, the Reviewer points out that local differences in F-actin dynamics might be expected in as complex an entity as a cell. We note that statistical models are broadly deployed in describing phenomena driven by many hidden variables that nonetheless converge on an emergent, statistically described ensemble behavior. Models for stochastic protein transcription/translation, for example, do not attempt to capture all the molecular details that control these processes. However, we do agree that the limitations of the model need to be made clear. We explicitly discuss one case where the model may fail, namely the lamellipodium, where directional actin flow is expected (see quote above). In addition, we note in the Discussion the possibility that the jump model may implicitly capture local heterogeneities in space and time:

“It is perhaps surprising that the jump model, or any model, fits the data given the compositional complexity of the cytoskeleton. It is likely that the best fit characteristic times, distances and reversal probabilities emerge from a distribution of timescales (and possibly length scales) that might be expected from the myriad of actin-binding proteins with varied kinetics, and that may be characterized by localized heterogeneities in space and time. Whether these emergent parameters arise in some limiting manner (e.g. due to central limit or extreme value theorems) or instead have a more complex physical origin remains to be determined.”

Reviewer #3 (Remarks to the Author):

The manuscript is much improved in terms of accessibility, advance in understanding the movement of actin, and methodology. The new findings on the jump model explaining the behavior of 4 difference actin structures and the clarified methods and model assumptions make this an interesting work worthy of publication.

We are gratified that the Reviewer finds the manuscript to be ready for publication. We very much appreciate the time and thought the Reviewer has put into improving our work.

The remaining concern is on significance. Statistical analysis on the differences between actin structures presented in Figure 4g should be done in order to determine the whether the differences have the potential to be biologically significant. Furthermore, the reasoning of the how the computed fit times, distances, and reversal probabilities for each actin structure might explain the cellular behavior of those structures should be explicitly described. The discussion speculates that these are at play, but assumes that the reader has the knowledge on hand to connect this speculation to each actin structure. Please add the rationale behind each parameter difference for each relevant actin structure to the discussion.

We apologize for this omission. To be conservative, we analyzed each biological replicate separately, and then derived test statistics across replicates using one-way

ANOVA followed by Tukey's range test to identify statistically significant differences in means. These comparisons are now in Supplementary Table 3. We have also added an additional experimental replicate to the analysis, which we note was collected in the presence of 0.1% DMSO. There is no observable difference in cell phenotype or fit parameters in comparing replicates with and without 0.1% DMSO. On the balance we feel that the inclusion of these additional data improves the robustness of our results, but we will of course revert to the earlier dataset if the Editor and Reviewer prefer.

We have expanded the Discussion exactly along the lines suggested by the Reviewer:

“Variations in the local density of actin crosslinkers and binding proteins may contribute to population changes in jump distance, as evidenced by the increase in jump size observed in cells treated with latrunculin A. In this regard it is intriguing that the magnitude of jump distances ~50 nm is similar to the actin network mesh size (10s-100s of nanometers) (Morone et al. 2006). Waiting times between jumps may reflect the balance of kinetics between force-generating myosin motors and structure-stabilizing actin crosslinkers. Times that span roughly 5 to 25 s are commensurate with the timescale of F-actin crosslinker rearrangement inferred in a recent study from our laboratory (Tan et al. 2020). The longer time between jumps in the stress fiber population may likewise reflect the greater stability of these structures, which have numerous cross-linking proteins in addition to force-generating myosins. At a lower temperature (26 °C), which is expected to slow molecular-level kinetics, the best fit characteristic time between jumps is increased, but both cortical and stress fiber populations exhibit similar average jump distances relative to those measured at 37 °C. Last, the probability of switching may reflect the inherent directionality of local forces. For example, the switching probability was lowest over focal adhesions, where there is consistent inward-oriented traction forces as well as local actin polymerization, both of which might be expected to yield directional F-actin movement.”

Reviewers' Comments:

Reviewer #2:

Remarks to the Author:

The authors removed previous Supplementary Figure 1 after I pointed out the time-dependent increase of the sigma values in the previous graphs. This is not a good behavior. Please do not delete the data.

As I thoroughly explained in my 1st review, the accuracy in both particle tracking and localization may rapidly become erratic when the particle density is increased to a certain level (Nature Methods 11, 281, 2014, Traffic 18, 840, 2017, now I add another comprehensive study, Nature Methods 12, 717, 2015). The images in Fig 1a and 2G do not appear appropriate for single-molecule tracking analysis (by using any state-of-art software) because the label density is very high along stress fibers and focal adhesions. The limitation of this study is the lack of solutions for estimating potentially large fluctuations of sigma values between different cell locations and different time frames (the latter was highly suggested by the previous Suppl Fig. 1 data even now. If focus or stage drift was the reason for the sigma increase, the live cell data were also affected. In addition, the authors should write their arguments in the paper not in the rebuttal). Because the lack of careful response to my previous criticisms by the authors, I do not recommend publication in Nature Communications.

Reviewer #3:

Remarks to the Author:

My concerns have been satisfactorily addressed.

The more thorough discussion of position vs. velocity jump models is much appreciated, as well as the new paragraph that contextualizes the potential significance of their findings. Figure 4C and 4D are clear.

Minor comments:

-The explanation for Fig 4 that "relatively rapid and heterogeneous movements at the single-filament level, averaged over many filaments and multiple filament lifetimes, are expected to yield the ensemble-level movements seen previously [in bulk measurements of continuous behavior]." could be added to the paper proper.

-Typo "Fig. xx" – line 359

-Typo "stress fiber F-actin, an[d] observation that is consistent with" – line 413

The authors removed previous Supplementary Figure 1 after I pointed out the time-dependent increase of the sigma values in the previous graphs. This is not a good behavior. Please do not delete the data.

We sincerely apologize for any offense inadvertently caused. So far as we can tell, concerns on this point arise from a misunderstanding. The purpose of this measurement is to determine the accuracy with which we can assign the locations of presumably stationary puncta in fixed cells, which in turn provides an estimate of our localization accuracy. One way of doing so is to fit the fluorescent counts arising from a given punctum to a double Gaussian or analogous function, and then track the center through time. In the alternative approach we happened to take, we noted that the distribution of distances measured for pairs of localizations for the same fluorophore at two times t_1 and t_n is described by a noncentral distribution with 2 degrees of freedom, which yields an equivalent estimation of the localization uncertainty σ :

$$f_d(d, s, \sigma) = \frac{d}{2\sigma^2} \exp\left(-\frac{1}{4\sigma^2}(d^2 + s^2)\right) I_0\left(\frac{ds}{2\sigma^2}\right),$$

where d is the measured distance between two points, s is the true distance (here, zero), and I_0 is a modified Bessel function of the first kind.

In the initial draft we had reported that an *apparent* σ increased with the time separation between frames. After the questions regarding this in the second round of review, we tracked down the origin of this effect and found it was likely due to residual sample drift. Accurate values of σ are easily found using time points that are sufficiently close together such that residual drift is not an issue. The updated Figure S1 directly quantifies residual sample drift, which is more, not less, transparent to the reader than the original version.

So, to summarize: σ is *not* time dependent, or at least not in a way we have been able to discern. (This is investigated in further detail below.) However, our initial strategy for estimating σ showed an *apparent* time-dependence, which we subsequently tracked down as stemming from sample drift. We hope that in explaining this the reviewer will be satisfied both with the measurement and that we are following good scientific practices. All that said, we are happy to additionally include the original version of S1 if the reviewer thinks it will improve the manuscript.

As I thoroughly explained in my 1st review, the accuracy in both particle tracking and localization may rapidly become erratic when the particle density is increased to a certain level (Nature Methods 11, 281, 2014, Traffic 18, 840, 2017, now I add another comprehensive study, Nature Methods 12, 717, 2015). The images in Fig 1a and 2G do not appear appropriate for single-molecule tracking analysis (by using any state-of-art software) because the label density is very high along stress fibers and focal adhesions. The limitation of this study is the lack of solutions for estimating potentially large fluctuations of sigma values between different cell locations and different time frames (the latter was highly suggested by the previous Suppl Fig. 1 data even now. If focus or stage drift was the reason for the sigma increase, the live cell data

were also affected. In addition, the authors should write their arguments in the paper not in the rebuttal).

We wish to assure the reviewer that we took his/her criticisms very much to heart, and have endeavored throughout to address them. In fact, we had carefully considered this potential issue from the inception of the project, as it is a possibility that would occur to anyone with even a casual familiarity with the field. In our study, we are interested in the aggregate, statistical properties that describe the motion of F-actin filaments. Thus, we do not aim to track every fluorophore in a field of view. Instead, it is important that the subset of filaments that we do analyze are indeed accurately tracked.

As we explained in the first round of review, given the aims of our study, we carefully exclude puncta that cannot be reliably localized, and instead limit our analysis to the subset of molecules that can be reliably tracked (as described in Materials and Methods, Actin tracking analysis).

We begin by tracking particles with QFSM (Mendoza et al. *Current Protocols in Cytometry* 2012). QFSM uses iterative speckle detection to more accurately extract speckle localizations in crowded environments, and leverages the tracking algorithm described in Jaqaman et al. *Nature Methods* 2008, which was designed to improve tracking at higher densities and to account for heterogeneity in particle movements.

We do not include all localizations detected by QFSM in our analysis, however. These are subjected to an additional layer of filtering: localizations that cannot be fit to a Gaussian in x and y with reasonable parameters (best fit center within 2 pixels (128 nm) of the position given by QFSM, standard deviation within a factor of 1.5 of $0.25\lambda/NA$ ($\lambda = 675$ nm, $NA = 1.49$)) are discarded. As a result, less than half of puncta initially identified by QFSM are retained for analysis. As might be expected, fluorophores that are rejected in this second step tend to be dim and/or close to a neighboring fluorophore. A useful illustration is provided by **Supplemental Movie 2**. This movie also shows that, in this hybrid analysis, puncta in initially dense regions of the cell, for example stress fibers, are for the most part not tracked in the initial frames of the movie. Instead, these regions yield a larger number of trackable puncta as photobleaching decreases the overall density of fluorophores.

To this last point, both Figure 1A and Figure 2G show the first frames of the corresponding datasets. The number (and density) of puncta drops rapidly as the movie progresses, as the reviewer can confirm by watching Movies S1 and S2 (see also point 1, below).

Although we were confident in our basic approach, we did take this critique quite seriously. In the initial round of review, we included extensive new control measurements to demonstrate the robustness of the method. These were described in detail in the first rebuttal letter, so we will summarize here:

- 1) In the previous version of Figure S12, we compare displacement distributions measured for the first 10 and last 10 frames for live-cell movies, at which point fluorophore density was

decreased by >50% due to photobleaching. If there were a problem with tracking arising from overlapping puncta, it should show up in this comparison. In fact, displacement distributions measured over both time windows are effectively identical. If either localization or tracking error were significantly altering our measurements, one would expect the two displacement distributions to differ significantly from one another.

For completeness, in the new revision we repeated this measurement across multiple time windows (i.e. 0-10, 10-20, 20-30 s, etc.) (previous Figure S12, modified figure is now Figure S12b). Consistent with expectation, we see the same displacement distribution across all these time frames. In addition, we repeat this calculation but for 2 s between frames, when the displacements of the large majority of puncta are anticipated to be small relative to localization error (new Fig. S12a). While the 2 s is dominated by localization error, the 10 s measurement includes a substantial contribution from true, underlying movement. The fact that distributions on both timescales are constant across time indicates that neither the localization error nor the true displacements are time-dependent.

Supplementary Figure 12. Measured actin speckle displacements over varying portions of a movie. Displacements over a 2 s time interval (a) and a 10 s time interval (b) in live cells (all subcellular populations), measured during varying portions of a movie (frames 1-10, 11-20, ..., 51-60). $n = 9$ cells.

2) In Figure S13, we compare velocity fit parameters from the dimmest 25%, brightest 25%, and middle 50% of puncta. Despite higher error in the localizations from dimmer puncta, the three populations yield similar fit parameters, indicating that local differences in localization error do not significantly affect our best fit velocity distributions.

3) To test for tracking errors, we compare fit parameters from all tracks to fit parameters from long (>16 s) tracks (Table S2). One would expect, in the presence of significant tracking errors, a significantly different outcome from these fits. While the precise numbers are not identical, both fits show the same trends and support the main claims of the paper: namely that a statistical jump model fits the data better than a diffusive model, and that the best fit parameters

are consistent with more frequent and farther jumps in the cortical population than in stress fibers.

4) Data collected in a second cell type, HUVECs, yielded the same qualitative results (Figure S8), despite a lower effective labeling density in these cells. We have updated Supplementary Figure 8 to show the labeling density in panel a, shown below, and added an additional supplementary video (S3), showing an acquisition of the SiR-Actin signal in a live HUVEC over time. Again, one would not expect the outcomes of experiments at two labeling densities to be similar if there was significant error arising from puncta density.

Supplementary Figure 8, panel a. Representative images of two HUVECs, each shown in brightfield (left) and in the SiR-actin channel (right), after denoising with noise2void. Panels of zoomed in regions show the labeling density at the beginning of the acquisition.

5) Although it is not direct evidence that our tracking is accurate, model fit parameters change in ways that are biologically reasonable for perturbations such as changes in temperature (Supplementary Figure 9) and cytoskeletal inhibitors (Supplementary Figure 10), and differ in sensible ways when comparing stress fibers and the cortex (Figure 4g).

As there was no comment (positive or negative) on these data in the second round of review, we concluded at the time, perhaps erroneously, that this concern had been adequately addressed.

Nevertheless it is good that the point was brought up again, as it has caused us to perform an additional analysis that we think will further strengthen the manuscript. To address whether there is spatial variation in localization error, we have plotted σ values for individual puncta in fixed cells, as shown in Figure S14 and reproduced below:

Supplementary Figure 14 Three representative fixed cells, where each track with at least 5 subpixel localizations is shown. Tracks are represented by circles centered at their mean positions, and are color-coded by the standard deviations of their positions (averaged over x and y for each).

These images show that there is no discernable spatial pattern associated with σ values, either with respect to subcellular localization or within or outside of stress fibers.

To summarize, we find no evidence that the true, physical value of σ varies as a function of time or subcellular location for live-cell data. Further, we have checked in multiple ways to make sure that particle tracking accuracy is not a limiting factor in our analysis.

Regarding information in the rebuttal letter vs. the SI, we have carefully checked that all the information contained in response letters is directly accessible to readers. In so doing, we have added a new Figure S16, showing example actin tracks, that was previously included only in a response letter. We have also added the following passage to the Discussion, which recapitulates a clarification previously advanced in a response letter:

“While F-actin motion often resembles steady flow at the bulk level, it was unknown how the motion of individual filaments might be represented statistically. Like diffusion of particles down a concentration gradient, which has the visual effect of steady unidirectional movement, the paths of individual particles need not demonstrate steady forward motion. Here, we measured the motion of single puncta corresponding to individual actin filaments, and found these movements to be inconsistent with a model in which individual filaments move at a constant velocity (Figure 4, Supplementary Figure 16). Our findings are however consistent with previous studies whose results implied heterogeneity in the velocities of individual actin filaments (Yamashiro et al. MBoC 2014), and in particular that a subset of F-actin filaments should have zero actin velocities (Tan *et al.* Science Advances 2020; Driscoll *et al.* PNAS 2020). Thus, single-molecule and bulk measurements are not inconsistent: relatively rapid and heterogeneous movements at the single-filament level, averaged over many filaments and multiple filament lifetimes, are expected to yield the ensemble-level movements seen previously.”

In considering textural additions, we have tried to balance thoroughness with readability. If the reviewer or editor has suggestions for further expansions to the main text or supplemental information we will be happy to implement them.

Reviewer #3 (Remarks to the Author):

My concerns have been satisfactorily addressed.

The more thorough discussion of position vs. velocity jump models is much appreciated, as well as the new paragraph that contextualizes the potential significance of their findings. Figure 4C and 4D are clear.

We are gratified that the reviewer finds our work to be substantially ready for publication, and thank them for their time in critically evaluating our work.

Minor comments:

-The explanation for Fig 4 that “relatively rapid and heterogeneous movements at the single-filament level, averaged over many filaments and multiple filament lifetimes, are expected to yield the ensemble-level movements seen previously [in bulk measurements of continuous behavior].” could be added to the paper proper.

We have added a somewhat expanded version of this passage to the manuscript (please see the response to Reviewer 2, immediately above).

-Typo “Fig. xx” – line 359

-Typo “stress fiber F-actin, an[d] observation that is consistent with” – line 413

Thank you for catching these errors, they have been corrected.